



**Millennial-scale variations of sedimentary oxygenation in the western subtropical North Pacific and its links to the North Atlantic climate**

Jianjun Zou[1,2], Xuefa Shi[1,2], Aimei Zhu[1], Selvaraj Kandasamy[3], Xun Gong[4], Lester Lembke-Jene[4], Min-Te Chen[5], Yonghua Wu[1,2], Shulan Ge[1,2], Yanguang Liu[1,2], Xinru Xue[1], Gerrit Lohmann[4], Ralf Tiedemann[4]

[1]Key Laboratory of Marine Sedimentology and Environmental Geology, First Institute of Oceanography, MNR, Qingdao 266061, China

[2]Laboratory for Marine Geology, Qingdao National Laboratory for Marine Science and Technology, Qingdao, 266061, China

[3]Department of Geological Oceanography and State Key Laboratory of Marine Environmental Science, Xiamen University, Xiamen361102, China

[4]Alfred-Wegener-Institut Helmholtz-Zentrum für Polar- und Meeresforschung, Bussestr. 24, 27570 Bremerhaven, Germany

[5]Institute of Applied Geosciences, National Taiwan Ocean University, Keelung 20224, Taiwan

Corresponding authors:

Jianjun Zou (zoujianjun@fio.org.cn); Xuefa Shi (xfshi@fio.org.cn)

**Key Points**

1. History of sedimentary oxygenation processes at mid-depths in the western subtropical North Pacific since the Last Glacial is reconstructed using sediment-bound geochemical proxies.

2. Redox-sensitive proxies reveal millennial-scale variations in sedimentary oxygenation that correlated closely to changes in the North Pacific Intermediate Water.

3. A millennial-scale out-of-phase relationship between deglacial ventilation in the western subtropical North Pacific and the formation of North Atlantic Deep Water is suggested.



4. A larger $CO_2$ storage at mid-depths of the North Pacific corresponds to the
termination of atmospheric $CO_2$ rise during the Bölling-Alleröd interval.



**Abstract**

Lower glacial atmospheric $CO_2$ concentrations have been attributed to carbon

sequestration in deep oceans. However, potential roles of voluminous subtropical
North Pacific in modulating atmospheric $CO_2$ levels on millennial timescale are
poorly constrained. Further, an increase in respired $CO_2$ concentration in the glacial
deep ocean due to biological pump generally is coeval with less oxygenation in the
subsurface layer. This link thus offers a chance to visit oceanic ventilation and the
coeval export productivity based on redox-controlled sedimentary geochemical
parameters. Here we investigate a suite of sediment geochemical proxies to
understand the sedimentary oxygenation variations in the subtropical North Pacific
(core CSH1) over the last 50 thousand years (ka). Our results suggest that sedimentary
oxygenation at mid-depths of the subtropical North Pacific intensifies during the
episodes of late glacial (50-25 ka), Last Glacial Maximum (LGM) and also the
interval after 8.5 ka, especially pronounced for the North Atlantic millennial-scale
abrupt cold events of the Younger Dryas, Heinrich Stadial (HS) 1 and 2. On the other
hand, oxygen-depleted seawater is found during the Bölling - Alleröd (B/A) and
Preboreal. Our findings of enhanced sedimentary oxygenation in the subtropical North
Pacific is aligned with intensified formation of North Pacific Intermediate Water
(NPIW) during cold spells, while the ameliorated sedimentary oxygenation seems to
be linked with the intensified Kuroshio Current since 8.5 ka. In our results,
diminished sedimentary oxygenation during the B/A indicates an enhanced $CO_2$
sequestration at mid-depth waters, along with slight increase in atmospheric $CO_2$
concentration. Mechanistically, we speculate that these millennial-scale changes were
linked to the strength of North Atlantic Deep Water, leading to intensification of
NPIW formation and enhanced abyss flushing during deglacial cold and warm
intervals, respectively. Enhanced formation of NPIW seem to be driven by the
perturbation of sea ice formation and sea surface salinity oscillation in high latitude
North Pacific through atmospheric and oceanic teleconnection. During the B/A,
decreased sedimentary oxygenation likely resulted from an upward penetration of
aged deep water into the intermediate-depth in the North Pacific, corresponding to a



resumption of Atlantic Meridional Overturning Circulation.
**Keywords:** sedimentary oxygenation; millennial timescale; North Pacific
Intermediate Water; Atlantic Meridional Overturning Circulation; subtropical North
Pacific



## 1. Introduction


The sluggish ocean ventilation depletes dissolved oxygen in the sediment-water
interface and facilitates the storage of respired carbon, which in turn plays a role in
regulating sedimentary oxygen, potentially linking to atmospheric $CO_2$ changes on
orbital and millennial timescales (Jaccard and Galbraith, 2012; Lu et al., 2016;
Sigman and Boyle, 2000). The reconstruction of past sedimentary oxygenation is
therefore crucial for understanding changes in export productivity and the renewal
rate of deep ocean circulation (Nameroff et al., 2004). Previous studies from
high-latitude North Pacific margins and subarctic Pacific have identified drastic
variations in export productivity and marine oxygen levels at glacial-interglacial
timescales using diverse proxies such as trace elements (Cartapanis et al., 2011;
Chang et al., 2014; Jaccard et al., 2009; Zou et al., 2012), benthic foraminiferal
assemblages (Ohkushi et al., 2016; Ohkushi et al., 2013; Shibahara et al., 2007)and
nitrogen isotopic composition ($\delta^{15}N$) of organic matter (Chang et al., 2014; Galbraith
et al., 2004; Riethdorf et al., 2016) in cored sediments. These studies further
suggested that both North Pacific Intermediate Water (NPIW) and export of organic
carbon regulate the sedimentary oxygenation variation during the last glaciation and
Holocene in the northeast North Pacific. By contrast, little information exists on
millennial-scale oxygenation changes to date in the western subtropical North Pacific.
The modern NPIW is mainly sourced from the NW Pacific marginal seas
(Shcherbina et al., 2003; Talley, 1993; You et al., 2000), and then it spreads into
subtropical North Pacific at intermediate depths of 300 to 800 m (Talley, 1993). The
pathway and circulation of NPIW have been identified byYou (2003), who suggested
that cabbeling is the principle mechanism responsible for transforming subpolar
source waters into subtropical NPIW along the subarctic-tropical frontal zone. More
specifically, You et al. (2003) argued that a lower subpolar input of about 2Sv is
sufficient for subtropical ventilation. Benthic foraminiferal $\delta^{13}C$ data from the North
Pacific suggested enhanced ventilation at water depths of < 2000 m during the last
glacial period(Keigwin, 1998; Matsumoto et al., 2002). Furthermore, on the basis of
both radiocarbon data and modeling results, Okazaki et al. (2010) provided further





insight into the formation of deep water in the North Pacific during early deglaciation.
Enhanced NPIW penetration is further explored using numerical model simulations
(Chikamoto et al., 2012; Gong et al., 2019; Okazaki et al., 2010). The downstream
effects of intensified GNPIW can be seen in the record of $\delta^{13}$C of *Cibicides*
*wuellerstorfi* in core PN-3 from the middle Okinawa Trough (OT), whereas lower
deglacial $\delta^{13}$C values were attributed to enhanced OC accumulation rates due to
higher surface productivity by Wahyudi and Minagawa (1997).

The Okinawa Trough is separated from the Philippine Sea by the Ryukyu Islands

and is an important channel of the northern extension of the Kuroshio in the western
subtropical North Pacific (Figure 1). Surface hydrographic characteristics of the OT
over glacial-interglacial cycles are largely influenced by the Kuroshio and East China
Sea Coastal Water (Shi et al., 2014); the latter is related to the strength of summer
East Asian monsoon (EAM) sourced from the western tropical Pacific. Modern
physical oceanographic investigations showed that intermediate waters in the OT are
mainly derived from horizontal advection and mixture of NPIW and South China Sea
Intermediate Water (Nakamura et al., 2013). These waters intrude into the OT through
two ways (Nakamura et al., 2013): (i) deeper part of the Kuroshio enters the OT
through the channel east of Taiwan (sill depth 775 m) and (ii) through the Kerama
Gap (sill depth 1100 m). In the northern OT, the occupied subsurface water mainly
flows through the Kerama Gap through horizontal advection from the Philippine Sea
(Nakamura et al., 2013). Recently, Nishina et al. (2016) found that an overflow
through the Kerama Gap controls the modern deep-water ventilation in the southern
OT.

Both surface hydrography and deep ventilation in the OT varied greatly since the

last glaciation. During the last glacials, the mainstream of the Kuroshio likely
migrated to the east of the Ryukyu Islands or and also became weaker due to lower
sea levels (Shi et al., 2014; Ujiié and Ujiié, 1999; Ujiié et al., 2003) and the
hypothetical emergence of a Ryukyu-Taiwan land bridge (Ujiié and Ujiié, 1999). In a
recent study, based on the Mg/Ca-derived temperatures in surface and thermocline
waters and planktic foraminiferal indicators of water masses from two sediment cores

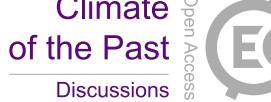

located in the northern and southern OT, Ujiié et al. (2016) further argued that the
hydrological conditions of North Pacific Subtropical Gyre since MIS 7 is modulated
by the interaction between the Kuroshio and the NPIW. Besides the Kuroshio, the
flux of East Asian rivers to the East China Sea (ECS), which is related to the summer
EAM and the sea level oscillation coupled with topography are also regulating the
surface hydrography in the Okinawa Trough (Chang et al., 2009; Kubota et al., 2010;
Sun et al., 2005; Yu et al., 2009).

Based on benthic foraminiferal assemblages, previous studies have implied a

reduced oxygenation in deep waters of the middle and southern OT during the last
deglacial period (Jian et al., 1996; Li et al., 2005), but a strong ventilation during the
Last Glacial Maximum (LGM) and the Holocene (Jian et al., 1996; Kao et al., 2005).
High sedimentary $\delta^{15}N$ values, an indicator of increased denitrification in the
subsurface water column, also occurred during the late deglaciation in the middle
OT(Kao et al., 2008). Inconsistent with these results, Dou et al. (2015) suggested an
oxic depositional environment during the last deglaciation in the southern OT based
on weak positive cerium anomalies. Furthermore, Kao et al. (2006) concluded a
reduced ventilation of deepwater in the OT during the LGM due to the reduction of
KC inflow using a 3-D ocean model. Yet, the patterns and reasons that caused
sedimentary oxygenation in the OT thus remain unclear.
**2. Paleo-redox proxies**

Sedimentary redox condition is the balance between the rate of oxygen supply

from the overlying bottom water and the rate of oxygen removal from pore water
(Jaccard et al., 2016), processes that are closely related to the advection of submarine
ocean circulation and organic matter respiration, respectively. Contrasting
geochemical behaviors of redox-sensitive trace metals (Mn, Mo, U, etc.) have been
extensively used to reconstruct bottom water and sedimentary oxygen changes (Algeo,
2004; Algeo and Lyons, 2006; Crusius et al., 1996; Dean et al., 1997; Tribovillard et
al., 2006; Zou et al., 2012), as their concentrations respond to redox condition of the
depositional environment (Morford and Emerson, 1999).

In general, enrichment of Mn with higher speciation states (Mn (III) and Mn (IV))



in the form of Mn-oxide coatings is observed in marine sediments, when oxic
condition prevails into great sediment depths as a result of low organic matter
degradation rates and well-ventilated bottom water (Burdige, 1993). In reducing
conditions, Mn is released as dissolved Mn (II) species into the pore water and thus its
concentration is usually low in suboxic ($O_2$ and $HS^-$ absent) and anoxic ($HS^-$ present)
sediments. In addition, when Mn enrichment occurs in oxic sediments as solid phase
Mn oxyhydroxides, it may lead to co-precipitation of other elements, such as Mo
(Nameroff et al., 2002).
The elements Mo and U behave conservatively in oxygenated seawater, but are
preferentially enriched in oxygen-depleted water. However, these two trace metals
behave differently in several ways. Molybdenum can be enriched in both oxic
sediments, such as the near surface manganese-rich horizons in continental margin
environments (Shimmield and Price, 1986) and in anoxic sediments (Nameroff et al.,
2002). Under anoxic conditions, Mo can be reduced either from the +6 oxidation state
to insoluble $MoS_2$, though this process is known to occur only under extremely
reducing conditions, such as hydrothermal and/or diagenesis (Dahl et al., 2010; Helz
et al., 1996) or be converted to particle-reactive thiomolybdates (Vorlicek and Helz,
2002). Zheng et al. (2000) suggested two critical thresholds for Mo scavenging from
seawater: 0.1µM hydrogen sulfide ($H_2S$) for Fe-S-Mo co-precipitation and 100 µM
$H_2S$ for Mo scavenging as Mo-S or as particle-bound Mo without Fe. Although
Crusius et al. (1996) noted insignificant enrichment of sedimentary Mo under suboxic
conditions, Scott et al. (2008) argued that burial flux of Mo is not so low in suboxic
environments. Excess concentration of Mo ($Mo_{excess}$) in sediments thus suggests the
accumulation of sediments either in anoxic ($H_2S$ occurrence) or well oxygenated
conditions (if $Mo_{excess}$ is in association with Mn-oxides).
In general, U is enriched in anoxic sediments (>1 µM $H_2S$), but not in oxic
sediments (>10 µM $O_2$) (Nameroff et al., 2002). Accumulation of U depends on the
content of reactive organic matter (Sundby et al., 2004) and U precipitates as uraninite
($UO_2$) during the conversion of Fe (III) to Fe (II) in suboxic conditions (Morford and
Emerson, 1999; Zheng et al., 2002). One of the primary removal mechanisms for U



from the ocean is via diffusion across the sediment-water interface of reducing
sediments (Klinkhammer and Palmer, 1991). Under suboxic conditions, soluble U (VI)
is reduced to insoluble U (IV), but free sulfide is not required for U
precipitation(McManus et al., 2005).Jaccard et al. (2009) suggested that the presence
of excess concentration of U ($U_{excess}$) in the absence of Mo enrichment is indicative of
a suboxic, but not sulfidic condition, within the diffusional range of the
sediment-water interface. The felsic volcanic is also a primary source of uranium
(Maithani and Srinivasan, 2011). Therefore, the potential input of uranium from active
volcanic sources around the northwestern Pacific to the adjacent sediments should not
be neglected.
In this study, we investigate a suite of redox-sensitive elements and the ratio of
Mo/Mn along with productivity proxies from a sediment core retrieved from the
northern OT to reconstruct the sedimentary oxygenation in the western subtropical
North Pacific over the last 50ka. Based on that, we propose that multiple factors, such
as NPIW ventilation, the strength of the Kuroshio Current and export productivity,
control the bottom sedimentary oxygenation in the OT on millennial time scales since
the last glacial.

**3. Oceanographic setting**

The OT resulted from the collision of the Philippine and Eurasian plates and
initially opened at the middle Miocene (Sibuet et al., 1987). Since that time, the OT
has been a depositional center in the ECS and receives large sediment supplies from
nearby rivers (Chang et al., 2009). At present, water depth in the axial part of the OT
deepens from 500 m in the north to ~2700 m in the south.
Surface hydrographic characteristics of the OT are mainly controlled by the
warmer, more saline Kuroshio water and cooler, less saline Changjiang Diluted Water,
and the modern flow-path of the former is influenced by the bathymetry of the OT
(Figure 1a). The Kuroshio Current originates from the North Equatorial Current and
flows into the ECS from the Philippine Sea through the Suao-Yonaguni Depression.
In the northern OT, Tsushima Warm Current (TWC), a branch of the Kuroshio, flows
into the Japan Sea through the shallow Tsushima Strait. Volume transport of the



Kuroshio varies seasonally due to the influence of the EAM with a maximum of 24
Sv (1 Sv = $10^6$ m$^3$ /s) in summer and a minimum of 20 Sv in autumn across the east of
Taiwan (Qu and Lukas, 2003).

Figures 2a and b show the lower sea surface salinity (SSS) zone in summer in the

ECS migrates toward the east of OT, indicating enhanced impact of the Changjiang
discharge associated with summer EAM. The mean annual discharge of the
Changjiang is 0.028 Sv and ~80% of its total discharge is supplied to the ECS
(Ichikawa and Beardsley, 2002). In situ observational data of surface hydrography
along the ship track from Taiwan Strait to Korea Strait and around the entrance of the
Tsushima Strait in the northern part of the ECS show a lower SSS in summer and a
negative correlation between the Changjiang discharge and SSS in July (Delcroix and
Murtugudde, 2002). Lower SSS in summer than that in winter suggests stronger
effects of summer EAM on surface hydrography over the Kuroshio Current (Sun et al.,
2005). Consistently, previous studies from the Okinawa Trough reported such close
relationship between summer EAM and SSS back to the late Pleistocene (Chang et al.,
2009; Clemens et al., 2018; Kubota et al., 2010; Sun et al., 2005).

Despite the effects of EAM and the Kuroshio, evidence of geochemical tracers

(temperature, salinity, oxygen, nutrients and radiocarbon-$\Delta^{14}$C)collected during the
World Ocean Circulation Experiment (WOCE) Expeditions in the Pacific (transects
P24 and P03) favors the presence of low saline, nutrient-enriched intermediate and
deep waters (Talley, 2007). Dissolved oxygen content is <100 μM at water depths of
below 600 m in the OT along WOCE transects PC03 and PC24 (Talley, 2007).
Modern oceanographic observations at the Kerama Gap reveal that upwelling in the
OT is associated with the inflow of NPIW and studies using box model predicted that
overflow through the Kerama Gap is responsible for upwelling (3.8–7.6 × $10^{-6}$m s$^{-1}$)
(Nakamura et al., 2013; Nishina et al., 2016).
**4.    Materials and methods**
**4.1. Chronostratigraphy of core CSH1**

A 17.3 m long sediment core CSH1 (31° 13.7' N, 128° 43.4' E; water depth: 703

m) was collected from the northern OT, close to the main stream of TWC (Figure 1b)





and within the depth of NPIW (Figure 1c) using a piston corer during *Xiangyanghong*09 Cruise in 1998. This location is thus enabling us to reconstruct millennial changes in the properties of TWC and NPIW. The expedition was carried out by the First Institute of Oceanography, Ministry of Natural Resources of China. Core CSH1 mainly consists of clayey silt and silt with occurrence of plant debris at some depth intervals (Ge et al., 2007) (Figure 3a). In addition, three layers of volcanic ash were observed at depths of 74−106 cm, 782−794 cm, 1570−1602 cm and these three intervals can be correlated with well-known ash layers, Kikai-Akahoya (K-Ah; 7.3 ka), Aira-Tanzawa (AT; 29.24 ka) and Aso-4 (roughly around MIS 5a) (Machida, 1999), respectively. The core was split and sub-sampled at every 4 cm interval and then stored in China Ocean Sample Repository at 4 °C until analysis.

Previously, some paleoceanographic studies have been conducted and a set of data have been investigated for core CSH1, including the contents of planktic foraminifers as well as their carbon ($\delta^{13}C$) and oxygen isotope ($\delta^{18}O$) compositions (Shi et al., 2014), pollen (Chen et al., 2006), paleomagnetism (Ge et al., 2007) and $CaCO_3$ (Wu et al., 2004). An age model for this core has been constructed by using ten Accelerator Mass Spectrometry (AMS) $^{14}C$ dates and six oxygen isotope ($\delta^{18}O$) age control points. The whole 17.3 m core contains *ca.* 88 ka-long record of continuous sedimentation (Shi et al., 2014).

It is noteworthy that previous age control points with constant radiocarbon reservoir throughout core CSH1 are used to reveal orbital-scale Kuroshio variations (Shi et al., 2014), but insufficient to investigate millennial-scale climatic events. On the basis of original age model, a higher abundance of *Neogloboquadrina pachyderma* (*dextral*) that occurred during warmer intervals, such as the B/A, has been challenging to explain reasonably. On the other hand, paired measurements of $^{14}C/^{12}C$ and $^{230}Th$ ages from Hulu Cave stalagmites suggest magnetic field change has greatly contributed to high atmospheric $^{14}C/^{12}C$ values at HS4 and the YD (Cheng et al., 2018). Thus a constant reservoir age assumed when calibrating foraminiferal radiocarbon dates using CALIB 6 software and the Marine13 calibration dataset (Reimer et al., 2013)for Core CSH1 may cause large chronological uncertainties.



Here, we therefore recalibrated the radiocarbon dates using CALIB 7.04 software
with Marine 13 calibration dataset (Reimer et al., 2013). Moreover, on the basis of
significant correlation between planktic foraminiferal $\delta^{18}O$ and Chinese stalagmite
$\delta^{18}O$ (Cheng et al., 2016), a proxy of summer EAM related to SSS of ECS, we
re-established the age model for core CSH1 (Figures 3b-d). Overall, the new
chronological framework is similar tothe one previously reported by Shi et al. (2014),
but with more dates. In order to compare with published results associated with
ventilation changes in the North Pacific, here we mainly report the history of
sedimentary oxygenation in the northern OT since the last glacial. Linear
sedimentation rate varied between 10 and 60 cm/ka. The age control points were
shown in Table 2.
**4.2. Chemical analyses**
Sediment subsamples for geochemical analyses were freeze-dried and ground to
a fine powder with an agate mortar and pestle. Based on the age model, 85
subsamples from core CSH1 with time resolution of about 200 years (4 cm interval)
were selected for detailed geochemical analyses of major and minor elements and
total contents of carbon (TC), organic carbon (TOC) and nitrogen (TN). The
pretreatment of sediment and other analytical methods have been reported elsewhere
(Zou et al., 2012)
TC and TN were determined with an elemental analyzer (EA; Vario EL III,
Elementar Analysen systeme GmbH) in the Key Laboratory of Marine Sediment and
Environment Geology, First Institute of Oceanography, Ministry of Natural Resources
of China, Qingdao. Carbonate was removed from sediments by adding 1M HCl to the
homogenized sediments for total organic carbon (TOC) analysis using the same
equipment. The content of calcium carbonate ($CaCO_3$) was calculated using the
equation:
$CaCO_3 = (TC-TOC) \times 8.33$
where 8.33 is the ratio between the molecular weight of carbonate and the atomic
weight of carbon. National reference material (GSD-9), blank sample and replicated
samples were used to control the analytical process. The relative standard deviation of



the GSD-9 for TC, TN and TOC is ≤ 3.4%.
About 0.5 g of sediment powder was digested in double distilled $HF:HNO_3$ (3:1),
followed by concentrated $HClO_4$, and then re-dissolved in 5% $HNO_3$. Selected major
and minor elements such as aluminum (Al) and manganese (Mn) were determined by
inductively coupled plasma optical emission spectroscopy (ICP-OES; Thermo
Scientific iCAP 6000, Thermo Fisher Scientific), as detailed elsewhere (Zou et al.,
2012). In addition, Mo and U were analyzed with inductively coupled plasma mass
spectrometry (ICP-MS; Thermo Scientific XSERIES 2, Thermo Fisher Scientific), as
described in Zou et al. (2012). Precision for most elements in the reference material
GSD-9 is ≤ 5% relative standard deviation. The excess fractions of U and Mo were
estimated by normalization to Al:
Excess fraction = $total_{element}$— (element/$Al_{average\ shale}$×Al),with U/$Al_{average\ shale}$=
$0.307×10^{-6}$ and Mo/$Al_{average\ shale}$=$0.295×10^{-6}$ (Li and Schoonmaker, 2014).
In addition, given the different geochemical behaviors of Mn and Mo and
co-precipitation and adsorption processes associated with the redox cycling of Mn, we
calculated the ratio of Mo to Mn, given that higher Mo/Mn ratio indicates lower
oxygen content in the depositional environment and vice versa. In combination with
the concentration of excess uranium, we infer the history of sedimentary oxygenation
in the subtropical North Pacific since the last glaciation.
**5.   Results**
**5.1. TOC, TN, and CaCO$_3$**
TN content shows a larger variation compared to TOC (Figure 4b), but it still
strongly correlates with TOC (r = 0.74, p<0.01) throughout the entire core.
Concentration of TOC ranges from 0.5 to 2.1% and it shows higher values with stable
trends during the last glacial phase (MIS 3) (Figure 4c). Molar ratios of TOC/TN vary
around 10, with higher ratios at the transition into the LGM (Figure 4d),
corresponding to higher linear sedimentation rate (Figure 4a). The content of $CaCO_3$
varies from 8.8 to 35% (Figure 4e) and it mostly shows higher values with increasing
trends during the last deglaciation. Conversely, the content of $CaCO_3$ is low and
exhibits decreasing trends during late MIS 3 and the LGM (Figure 4e).



Both TOC and CaCO$_3$ are widely used as proxies for the reconstruction of past
export productivity (Rühlemann et al., 1999). Molar C/N ratios of >10 (Figure 4c)
suggest that terrigenous organic sources significantly contribute to the TOC
concentration in core CSH1. The TOC content therefore may be not a reliable proxy
for the reconstruction of surface water export productivity during times of the LGM
and late deglaciation, when maxima in C/N ratios co-occur with decoupled trends
between CaCO$_3$ and TOC concentrations. In addition, the negative correlation
coefficient (r = – 0.85, p<0.01) between Al and CaCO$_3$ in sediments throughout core
CSH1 confirms the biogenic origin of CaCO$_3$ against terrigenous Al (Figure 4f).
Moreover, a detailed comparison between CaCO$_3$ concentrations and the previously
published foraminiferal fragmentation ratio (Wu et al., 2004) clearly shows, apart
from a small portion within the LGM, no clear co-variation. This suggests that CaCO$_3$
changes are primarily driven by variations in carbonate primary production, and not
overprinted by secondary processes, such as carbonate dissolution through changes in
the lysocline depth. On the other hand, terrigenous dilution generally decreases the
content of CaCO$_3$. The increasing trend of CaCO$_3$ associated with high sedimentation
rate (Figures 4a, e) indicates a substantial increase in export productivity during the
last deglacial interval. Thus, we can confidently use CaCO$_3$ content as productivity
proxy to a first order approximation.
**5.2. Redox-sensitive Elements**
Figure 4 shows time series of selected redox-sensitive elements (RSEs) and
proxies derived from them. Mn shows higher concentrations during the LGM and
HS1 (16–22.5ka) and middle-late Holocene, but lower concentrations during the last
deglacial and Preboreal periods (15.8–9.5ka) (Figure 4g). Generally, concentrations of
excess Mo and excess U (Figures 4j, l) show coherent patterns with those of Mo and
U (Figures 4i, k), but both are out-of-phase with Mn over the last glacial period
(Figure 4h). It should be noted that pronounced variations in U concentration since
8.5 ka is related to the occurrence of discrete volcanic materials. A negative
correlation between Mn and Mo$_{excess}$ during the last glaciation and the Holocene, and
the strong positive correlation between them during the LGM and HS1 (Figures 5a





and 5b) further corroborate the complicated geochemical behavior of Mn and Mo. A
strong positive correlation between $Mo_{excess}$ and Mn (Figure 5b) seems to be attributed
to co-precipitation of Mo by Mn-oxyhydroxide under oxygenated conditions. Here,
we use Mo/Mn ratio, instead of excess Mo concentration to reconstruct variations in
sedimentary redox state. Overall, the Mo/Mn ratio shows similar downcore pattern to
that of $Mo_{excess}$ with higher values during the last deglaciation, but lower values
during the LGM and HS1. A strong correlation between Mo/Mn ratio and excess U
concentration (Figure 5c) further corroborates the integrity of Mo/Mn as an indicator
of sedimentary oxygenation changes.
Rapidly decreasing Mo/Mn ratio indicates an oxygenated sedimentary
environment since ~8 ka (Figure 4h). Both higher Mo/Mn ratios and excess U
concentration, together with lower Mn concentrations suggest an oxygen-deficient
sedimentary environment during the late deglacial period (15.8–9.5 ka), whereas
lower values during the LGM, HS1 and HS2 indicate relatively better oxygenated
sedimentary condition. A decreasing trend in Mo/Mn ratio and excess U concentration
from 50 ka to 25 ka also suggest higher sedimentary oxygen levels.
**6.Discussion**
**6.1. Constraining paleoredox conditions in the Okinawa Trough**
In general, three different terms, hypoxia, suboxia and anoxia, are widely used to
describe the degree of oxygen depletion in the marine environment (Hofmann et al.,
2011). Generally, redox states in waters can be classified as oxic (>89 μmol /L $O_2$),
suboxic (~8.9-89 μmol/L$O_2$), anoxic–nonsulfidic (<89 μmol/L $O_2$, 0 μmol/L $H_2S$),
and anoxic– sulfidic or euxinic (0 ml $O_2$/L, >0 μmol/L $H_2S$) (Savrda and Bottjer,

1991).

Proxies associated with RSEs, such as sedimentary Mo concentration (Lyons et
al., 2009; Scott et al., 2008) have been used to constrain the degree of oxygenation in
seawater. Algeo and Tribovillard (2009) proposed that open-ocean systems with
suboxic waters tend to yield $U_{excess}$ enrichment relative to $Mo_{excess}$ and to result in
sediment $(Mo/U)_{excess}$ ratio less than that of seawater (7.5-7.9). Under increasingly
reducing and occasionally sulfidic conditions, the accumulation of $Mo_{excess}$ increase



relative to that of $U_{excess}$ leading the $(Mo/U)_{excess}$ ratio either is equal to or exceeds
with that of seawater. Furthermore, Scott and Lyons (2012) suggested a non-euxinic
condition with the presence of sulfide in pore waters, when Mo concentrations range
from> 2 ppm, the crustal average to < 25 ppm, a threshold concentration for euxinic
condition. Given that the Okinawa Trough is located in weakly restricted basin
settings, we use these two above mentioned proxies to evaluate the degree of
oxygenation in sediments.
Both bulk Mo concentration (1.2-9.5 ppm) and excess (Mo/U) ratio (0.2-5.7) in
core CSH1 suggest that oxygen-depleted conditions may have prevailed in the deep
water of the northern OT over the last 50 ka. However, increased excess Mo
concentration with enhanced Mo/U ratio during the last termination (18-9 ka) indicate
a stronger reducing condition compared to the Holocene and the last glacial period,
though Mo concentration is less than 25 ppm, a threshold for euxinic deposition
proposed by Scott and Lyons (2012).
The relative abundance of benthic foraminiferal species that thrive in different
oxygen concentrations also have been widely used to reconstruct the variations in
bottom water ventilation, such as enhanced abundance *of Bulimina aculeata*,
*Uvigerina peregrina and Chilostomella oolina* found under oxygen-depleted
conditions in the central and southern OT during the last deglaciation (Jian et al., 1996;
Li et al., 2005). An oxygenated bottom water condition is also indicated by abundant
benthic foraminifera species *Cibicidoides hyalina* and *Globocassidulina subglobosa*
(Jian et al., 1996; Li et al., 2005) and high benthic $\delta^{13}C$ values (Wahyudi and
Minagawa, 1997) in cores E017 (water depth 1826 m) , 225 (water depth 1575 m) and
PN-3 (water depth 1058 m) from the middle and southern OT (Figures 1 and 3)
during the postglacial period. The inferred ventilation pattern from benthic
foraminiferal assemblages is consistent with the one inferred from RSEs in this study
(Figure 4). Although we did not carry out benthic foraminiferal species analyses for
our core CSH1, it is reasonable to infer based on RSEs that the deepwater in the
northern OT was also in a prominent oxygen-poor condition during the late deglacial
interval. A clear link thus can be built between the ventilation of deep water and the



sedimentary oxygenation in the OT. In brief, our proxy records of RSEs in core CSH1
show oxygen-rich conditions during the last glaciation and middle and late Holocene
(since 8.5 ka) intervals, but oxygen-poor conditions during the late deglacial period.
**6.2. Causes for sedimentary oxygenation variations**
As discussed above, the pattern of RSEs in core CSH1 suggest that drastic
changes in sedimentary oxygenation occurred on orbital and millennial timescales
over the last glaciation in the Okinawa Trough. In general, four factors can regulate
the redox condition in the deep water column: (i) $O_2$ solubility, (ii) export productivity
and subsequent degradation of organic matter, (iii) vertical mixing, and (iv) lateral
provision of oxygen through intermediate and deeper water masses (Ivanochko and
Pedersen, 2004; Jaccard and Galbraith, 2012). In the OT, the oxygen deficiency
during the late deglacial period can be caused either by one and/or a combination of
more than one of these factors. In order to uncover the mechanisms responsible for
sedimentary oxygenation variations in the basin-wide OT, we compare our proxy
records of sedimentary oxygenation ($U_{excess}$ concentration and Mo/Mn ratio) and
export productivity ($CaCO_3$) with benthic foraminifera $\delta^{13}C$, a proxy for water mass,
in core PC23A (Rella et al., 2012), abundance of benthic foraminifera in the NE
Pacific, an indicator of anoxic condition (Cannariato and Kennett, 1999), abundance
of *Pulleniatina obliquiloculata*, an indicator of the Kuroshio strength (Shi et al., 2014)
(Figure 6).
**6.2.1. Effects of regional ocean temperature on deglacial deoxygenation**
Warming ocean temperatures lead to lower oxygen solubility. In the geological
past, solubility effects connected to temperature changes of the water column thought
to enhance or even trigger hypoxia (Praetorius et al., 2015). For instance, Shi et al.
(2014) reported an increase in SST of around 4°C (~21°C to ~24.6°C) during the last
deglaciation in core CSH1. Based on thermal solubility effects, a hypothetical
warming of 1°C at our site would reduce oxygen concentrations by about 8 μM
(Benson and Krause, 1984), which could drive a drastic drop of oxygen concentration
by <30 μM in subsurface water of the OT. Therefore, we assume that the late
deglacial hypoxia in the OT underwent a similar increase in ocean temperatures.



However, given the semi-quantitative nature of our data about oxygenation changes,
which seemingly exceed an amplitude of >30 μM, we suggest that other factors, in
particular processes like local changes in export productivity, regional influences such
as vertical mixing due to changes of the Kuroshio Current, as well as far-field effects
all may have played some decisive roles in shaping the oxygenation history of the OT.
**6.2.2. Links between deglacial primary productivity and sedimentary**
**deoxygenation**
Previous studies have suggested the occurrence of high primary productivity in
the entire OT during the last deglacial period (Chang et al., 2009; Jian et al., 1996;
Kao et al., 2008; Li et al., 2017; Shao et al., 2016; Wahyudi and Minagawa, 1997).
Such an increase in export production was due to favorable conditions for bloom
development, which were likely induced by warm temperatures and maxima in
nutrient availability, the latter being mainly sourced from increased discharge of the
Changjiang River, erosion of material from the ongoing flooding of the shallow
continental shelf in the ECS, and upwelling of Kuroshio Intermediate Water (Chang
et al., 2009; Li et al., 2017; Shao et al., 2016; Wahyudi and Minagawa, 1997). On the
basis of sedimentary reactive phosphorus concentration, Li et al. (2017) concluded
that export productivity increased during warm episodes but decreased during cold
spells on millennial timescales over the last 91 ka in the OT. Gradually increasing
concentrations of $CaCO_3$ in core CSH1 during the deglaciation (Figure 6a) and little
changes in foraminiferal fragmentation ratios (Wu et al., 2004), are indicative of high
export productivity in the northern OT. Accordingly, our data indicate that an increase
in export productivity, which was previously reported from the middle and southern
OT during the last deglaciation, and thus was a pervasive, synchronous phenomenon
of entire study region at the outermost extension of the ECS.
As a consequence, high export productivity lowers oxygen concentrations in
deeper waters, due to subsurface consumption of oxygen by remineralization of
organic matter. Similar events of high export productivity have been extensively
reported in the entire North Pacific due to increased nutrient supply, high SST,
reduced sea ice cover, etc. (Crusius et al., 2004; Dean et al., 1997; Jaccard and



Galbraith, 2012; Kohfeld and Chase, 2011). In most of these cases, the increases in
productivity were likely also responsible for oxygen depletion in mid-depth waters,
due to exceptionally high oxygen consumption. However, the productivity changes
during the deglacial interval, very specifically $CaCO_3$, are not fully consistent with the
trends of excess U and Mo/Mn ratio (Figure 4). The sedimentary oxygenation thus
cannot be determined by export productivity alone.

### 495    6.2.3 Effects of the Kuroshio dynamics on sedimentary oxygenation

The Kuroshio Current, one of the main drivers of vertical mixing, has been

identified as the key factor in controlling modern deep ventilation in the Okinawa
Trough (Kao et al., 2006). However, the flow path of the Kuroshio in the Okinawa
Trough during the glacial interval remains a matter of debate. Planktic foraminiferal
assemblages in sediment cores from inside and outside the Okinawa Trough indicated
that the Kuroshio have migrated to the east of the Ryukyu Islands during the LGM
(Ujiié and Ujiié, 1999). Subsequently, Kao et al. (2006) based on modeling results
suggested that the Kuroshio still enters into the Okinawa Trough, but the volume
transport was reduced by 43% compared to the present-day transport and the outlet of
Kuroshio switches from the Tokara Strait to the Kerama Gap at -80 and -135m
lowered sea level. Combined with sea surface temperature (SST) records and ocean
model results, Lee et al. (2013) argued that there was little effect of deglacial sea-level
change on the path of the Kuroshio, which still exited the Okinawa Trough from the
Tokara Strait during the glacial period. Because the main stream of the Kuroshio
Current is at a water depth of ~150 m, the SST records are insufficient to decipher past
changes of the Kuroshio (Ujiié et al., 2016).On the other hand, low abundances of *P.*
*obliquiloculata* in core CSH1 in the northern OT (Figure 6e) indicate that the main
flow path of the Kuroshio may have migrated to the east of the Ryukyu Island (Shi et
al., 2014). Such a flow change would have been caused by the proposed block of the
Ryukyu-Taiwan land bridge by low sea level (Ujiié and Ujiié, 1999) and an overall
reduced Kuroshio intensity (Kao et al., 2006), effectively suppressing the effect of the
Kuroshio on deep ventilation in the OT. Our RSEs data show that oxygenated
sedimentary conditions were dominant in the northern OT throughout the last glacial





period (Figures 6a, b, c, d). The Kuroshio thus likely had a weak or even no effect on
the renewal of oxygen to the sedimentary environment during the last glacial period.
On the other hand, the gradually increasing abundance of *P.obliquiloculata*
(Figure 6e) from 15 ka onwards indicates an intensified Kuroshio Current. Matsumoto
et al. (2002) suggested that the influence of the present Kuroshio can reach to the
bottom depth of the permanent thermocline, which is approximately at 1000 m water
depth. However, as mentioned above, the effect of Kuroshio on the sedimentary
oxygenation was likely very limited during glacial period and only gradually
increasing throughout the last glacial termination. Therefore, while its effect on our
observed deglacial variation in oxygenation may provide a slowly changing
background condition in vertical mixing effects on the sedimentary oxygenation in the
OT, it cannot account for the first order, rapid oxygenation changes that we observe
between 18 and 9 ka, including indications for millennial-scale variations (Figure 6).
Better oxygenated sedimentary conditions since 8.5 ka coincided with intensified
Kuroshio (Shi et al., 2014; Ujiié et al., 2003), as indicated by rapidly increased SST
and *P. obliquiloculata* abundance in core CSH1 (Figure 6e). The re-entrance of the
Kuroshio into the OT (Shi et al., 2014) with rising eustatic sea level likely enhanced
the vertical mixing and exchange between bottom and surface waters, ventilating the
deep water in the OT. Previous comparative studies based on epibenthic $\delta^{13}$C values
indicated well-ventilated deep water feeding both inside the OT and outside off the
Ryukyu Islands during the Holocene (Kubota et al., 2010; Wahyudi and Minagawa,
1997). In summary, during the Holocene our observed enhanced sedimentary
oxygenation regime is mainly related to the intensified Kuroshio, while the effect of
the Kuroshio on OT oxygenation was limited before 15 ka.
**6.2.4. Effects of GNPIW on sedimentary oxygenation**
Relatively stronger oxygenated Glacial North Pacific Intermediate Water
(GNPIW), coined by (Matsumoto et al., 2002), has been widely documented in the
Bering Sea (Itaki et al., 2012; Kim et al., 2011; Rella et al., 2012), the Okhotsk Sea
(Itaki et al., 2008; Okazaki et al., 2014; Okazaki et al., 2006; Wang and Wang, 2008;
Wu et al., 2014), off east Japan (Shibahara et al., 2007), the eastern North Pacific



(Cartapanis et al., 2011; Ohkushi et al., 2013) and western subarctic Pacific (Keigwin,
1998; Matsumoto et al., 2002). The intensified ventilation of GNPIW is firstly
attributable to the displacement of formation source region to the Bering Sea
(Ohkushi et al., 2003) and then is further confirmed by Horikawa et al. (2010). Under
such conditions, the invasion of well-ventilated GNPIW into the OT through the
Kerama Gap would have replenished the water column oxygen in the OT, although
the penetration depth of GNPIW remains under debate(Jaccard and Galbraith, 2013;
Okazaki et al., 2010; Rae et al., 2014). Both a gradual decrease in excess U
concentration and an increase in Mo/Mn ratio during the last glacial period (25-50 ka)
validate such inference, suggesting pronounced effects of intensified GNPIW in the
OT.

During HS1, a stronger formation of GNPIW was recorded in the North Pacific

by a variety of studies. On the basis of paired benthic-planktic (B-P) $^{14}$C data and
model simulations, Okazaki et al. (2010) suggested that NPIW penetrated into a water
depth of ~2500 to 3000 m during HS1. In contrast, Max et al. (2014) argued against
deep water formation in the North Pacific and showed that GNPIW was
well-ventilated only to intermediate water depths (< 1400 m). Various mid- and
high-latitude North Pacific records of B-P $^{14}$C age offsets at the intermediate water
depth (<600–2000 m) showed an active production of GNPIW during HS1 (Max et al.,
2014; Sagawa and Ikehara, 2008). Moreover, Kubota et al. (2010) reported increased
subsurface water temperatures related to enhanced GNPIW contributions during HS1
at a water depth of 1166m (GH08, and young deep water was observed in the northern
South China Sea during HS1 (Wan and Jian, 2014).

All these multiple lines of evidence imply the presence of well-ventilated

intermediate water in the upper 2000 m of the North Pacific during HS1. At this point,
the effect of a strong GNPIW likely reached the South China Sea (Wan and Jian, 2014;
Zheng et al., 2016), further to the south the Okinawa Trough. The pathway of GNPIW
from numerical model simulations(Zheng et al., 2016) was similar to modern
observations (You, 2003). Thus, a persistent, cause and effect relation has been
established between GNPIW ventilation, the oxygen concentration of OT deepwater



and sediment redox state during HS1. In addition, our data also suggested a similarly
enhanced ventilation in HS2 (Figure 6) that must also be attributed to intensified
GNPIW.

Hypoxic conditions during Bölling-Alleröd (B/A) have been also widely

observed in the mid- and high-latitude North Pacific (Jaccard and Galbraith, 2012;
Praetorius et al., 2015). Our data, both excess U concentrations and Mo/Mn ratio
recorded in core CSH1 (Figures 6b-d), further reveal the expansion of
oxygen-depletion at mid-depth waters down to the subtropical NW Pacific during the
late deglacial period. Based on high relative abundances of radiolarian species,
indicators of upper intermediate water ventilation in core PC-23A, Itaki et al. (2012)
suggested that a presence of well-ventilated waters was limited to the upper
intermediate layer (200–500 m) in the Bering Sea during warm periods, such as the
B/A and Preboreal. Higher B-P foraminiferal $^{14}$C ages, together with increased
intermediate water temperature and salinity recorded in core GH02-1030 (off East
Japan) supported a weakened formation of NPIW during the B/A (Sagawa and
Ikehara, 2008).These lines of evidence indicate that the boundary between GNPIW
and North Pacific Deep Water shoaled during the B/A, in comparison to HS1. Based
on a comparison of two benthic foraminiferal oxygen and carbon isotope records from
off northern Japan and the southern Ryukyu Island, Kubota et al. (2010) found a
stronger influence of Pacific Deep Water on intermediate-water temperature and
ventilation at their southern than the northern locations, although both sites are
located at similar water depths (1166 m and 1212 m, respectively). Higher excess U
concentration and low Mo/Mn ratio in our core CSH1 during the B/A and Preboreal
suggest reduced sedimentary oxygenation, consistent with reduced ventilation of
GNPIW, contributing to the subsurface water suboxia in the OT.

During the YD, Mo/Mn ratio and excess U show a slightly decreased oxygen

condition in the northern OT. In contrast, benthic foraminiferal $\delta^{18}$O and $\delta^{13}$C values
in a sediment core collected from the Oyashio region suggested a strengthened
formation and ventilation of GNPIW during the YD (Ohkushi et al., 2016). This
pattern possibly indicates a time-dependent, varying contribution of distal GNPIW to



the deglacial OT oxygenation history, and we presume a more pronounced
contribution of organic matter degradation due to high export productivity during this
period, as suggested by increasing $CaCO_3$ content.

**6.3. Subtropical North Pacific ventilation links to North Atlantic Climate**

Our RSEs data show a substantial millennial variability in intermediate water
ventilation in the subtropical North Pacific. Notably, both enhanced ventilation during
HS1 and HS2 and oxygen-poor condition during the B/A respectively correspond to
the collapse and resumption of Atlantic meridional overturning circulation (AMOC)
(Bohm et al., 2015; McManus et al., 2004) (Figure 7 d). This is consistent with the
results of various modeling simulations (Chikamoto et al., 2012; Menviel et al., 2014;
Okazaki et al., 2010; Saenko et al., 2004), although these models had different
scenarios and causes for the observed effects in GNPIW formation, and ventilation
ages derived from B-P$^{14}$C (Freeman et al., 2015; Max et al., 2014; Okazaki et al.,
2012). These lines of evidence reveal a persistent link between the ventilation of
North Pacific and the North Atlantic climate. Such links have also been corroborated
by using proxy data and modeling experiment between AMOC and East Asian
monsoon during the 8.2 ka event (Liu et al., 2013), the Holocene (Wang et al., 2005)
and 34–60 ka (Sun et al., 2012). The mechanism linking East Asia with North
Atlantic has been attributed to an atmospheric teleconnection, such as the position and
strength of Westerly Jet and Mongolia-Siberian High (Porter and Zhisheng, 1995).
However, the mechanism behind such oceanic ventilation seesaw pattern between the
North Atlantic and North Pacific is still unclear.
Increased NPIW formation of HS1 may have been caused by enhanced
salinity-driven vertical mixing through higher meridional water mass transport from
the subtropical Pacific. Previous studies have proposed that intermediate water
formation in the North Pacific hinged on a basin-wide increase in sea surface salinity
driven by changes in strength of the summer EAM and the moisture transport from
the Atlantic to the Pacific (Emile-Geay et al., 2003). Several modeling studies found
that freshwater forcing in the North Atlantic could cause a widespread surface
salinification in the subtropical Pacific Ocean (Menviel et al., 2014; Okazaki et al.,



2010; Saenko et al., 2004). This idea has been tested by proxy data (Rodríguez-Sanz
et al., 2013; Sagawa and Ikehara, 2008), which indicated a weakened summer EAM
and reduced transport of moisture from Atlantic to Pacific through Panama Isthmus
owing to the southward displacement of ITCZ caused by a weakening of AMOC.
Along with this process, as predicted through a general circulation modeling, a
strengthened Pacific Meridional Overturning Circulation would have transported
more warm and salty subtropical water into the high-latitude North Pacific (Okazaki
et al., 2010). In accordance with comprehensive Mg/Ca ratio-based salinity
reconstructions, however, Riethdorf et al. (2013) found no clear evidence for such
higher salinity patterns in the subarctic northwest Pacific during HS1.

649   On the other hand, a weakened AMOC would deepen the wintertime Aleutian

Low (Okumura et al., 2009), which is closely related to the sea ice formation in the
marginal seas of the subarctic Pacific (Cavalieri and Parkinson, 1987). Intense brine
rejection, accompanied by expanded sea ice formation, would have enhanced the
NPIW formation. Recently our modeling-derived evidence suggests enhanced sea ice
coverage in the southern Okhotsk Sea and off East Kamchatka Peninsula (Gong et al.,
2019). In addition, higher advection of low-salinity water via the Alaskan Stream to
the subarctic NW Pacific was probably enhanced during HS1, related to a shift of the
Aleutian Low pressure system over the North Pacific, which could also increase sea
ice formation, brine rejection and thereafter intermediate water ventilation (Riethdorf
et al., 2013).

660   During the late deglaciation, ameliorating global climate conditions, such as

warming Northern Hemisphere, and a strengthened Asian summer monsoon, are a
result of changes in insolation forcing, greenhouse gases concentrations, and variable
strengths of the AMOC (Clark et al., 2012; Liu et al., 2009). During the B/A, a
decrease in sea ice extent and duration, as well as reduced advection of Alaska Stream
waters were indicated by combined reconstructions of SST and mixed layer
temperatures from the subarctic Pacific (Riethdorf et al., 2013). At that time, the
rising eustatic sea level (Spratt and Lisiecki, 2016) would have supported the
intrusion of Alaska Stream into the Bering Sea by deepening and opening glacial





closed straits of the Aleutian Islands chain, while reducing the advection of the Alaska
Stream to the subarctic Pacific gyre (Riethdorf et al., 2013). In this scenario, saltier
and more stratified surface water conditions would have inhibited brine rejection and
subsequent formation and ventilation of NPIW (Lam et al., 2013), leading to a
reorganization of the Pacific water mass, closely coupled to the collapse and
resumption modes of the AMOC during these two intervals.
**6.4 Increased storage of $CO_2$ at mid-depth water in the North Pacific at the B/A**

One of the striking features of RSEs data is higher Mo/Mn ratios and excess U

concentrations at the B/A, indicating a substantial oxygen-poor condition in the
subtropical North Pacific, coinciding with the termination of atmospheric $CO_2$
concentration rise (Marcott et al., 2014) (Figure 7a). As described above, it can be
related to the upwelling of nutrient- and $CO_2$-rich Pacific Deep Water due to
resumption of AMOC and enhanced export production. Although here we are unable
to distinguish these two reasons from each other, boron isotope data measured on
surface-dwelling foraminifera in core MD01-2416 situated in the western subarctic
North Pacific did reveal a decrease in near-surface pH and an increase in $pCO_2$ at this
time (Gray et al., 2018). That is to say, subarctic North Pacific is a source of relatively
high atmospheric $CO_2$ concentration at the B/A. Here we cannot conclude that the
same processes could have occurred in the subtropical North Pacific due to the lack of
well-known drivers to draw out of the old carbon in the deep sea into the atmosphere.
However, an expansion of oxygen-depletion zone in the entire North Pacific suggest
an increase in respired carbon storage at intermediate-depth in the subtropical North
Pacific, which likely stalls the rise of atmospheric $CO_2$. Our results support the
findings by Galbraith et al. (2007) and are consistent with the hypothesis of deglacial
flushing of respired carbon dioxide from an isolated, deep ocean reservoir(Marchitto
et al., 2007; Sigman and Boyle, 2000). Given the sizeable volume of the North Pacific,
potentially, once the respired carbon could be emitted to the atmosphere in stages,
which would play an important role in propelling the Earth out of the last ice age
(Jaccard and Galbraith, 2018).
**7. Conclusions**

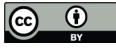



Our geochemical results revealed substantial changes in intermediate water redox
conditions in the northern Okinawa Trough over the last 50kaon orbital and millennial
timescales in the past. The sedimentary oxygenation variability presented here
provides key evidence for the impact of ventilation of NPIW on the sedimentary
oxygenation in the subtropical North Pacific and highlights the major role of Atlantic
Meridional Overturning Circulation in regulating the variations in sedimentary
oxygenation in the Okinawa Trough through ventilation of NPIW. Combined with
other published records, we also suggest an expansion of oxygen-depleted zone and
accumulation of respired carbon at the mid-depth waters of the North Pacific at the
B/A, coinciding with the termination of atmospheric $CO_2$ rise. Once the release of the
sequestered carbon into the atmosphere in stages, it would be helpful to maintain high
atmospheric $CO_2$ levels during the deglaciation and to propel the earth out of the
glacial climate.

**Data availability.** All raw data are available to all interested researchers upon request.

**Author Contributions.** J.J.Z. and X.F.S. conceived the study. A.M.Z. performed
geochemical analyses of bulk sediments. J.J.Z., X.F.S. K.S. and X.G. led the write up
of the manuscript. All other authors provided comments on the manuscript and
contributed to the final version of the manuscript.

**Competing interests:** The authors declare no competing interests.

**Acknowledgements**
Financial support was provided by the National Program on Global Change and
Air-Sea Interaction (GASI-GEOGE-04), by the National Natural Science Foundation
of China (Grant Nos.: 41476056, 41876065, 41420104005, 41206059, and U1606401)
and by the Basic Scientific Fund for National Public Research Institutes of China
(No.2016Q09) and International Cooperative Projects in Polar Study (201613)and
Taishan Scholars Program of Shandong.This study is a contribution to the bilateral



Sino-German collaboration project (funding through BMBF grant 03F0704A –
SIGEPAX). XG, LLJ, GL, RT thank the bilateral Sino-German collaboration
NOPAWAC project (BMBF grant No. 03F0785A).LLJ and RT acknowledge financial
support through the national Helmholtz REKLIM Initiative. We would like to thank
the anonymous reviewers, who helped to improve the quality of this manuscript. The
data used in this study are available from the authors upon request
(zoujianjun@fio.org.cn).

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





**Captions**

**Table 1.** Locations of different records and their source references discussed in the text.

**Table 2**. Age control points adopted between planktic $\delta^{18}$O of Core CSH1 and Chinese stalagmite $\delta^{18}$O (Cheng et al., 2016) for tuning the age model between 10 ka and 60 ka in this study. A linear interpolation was assumed between age control points.

**Figure 1.** (a) Spatial distribution of dissolved oxygen content at 700 m water depth in the North Pacific. Black arrows denote simplified Kuroshio and Oyashio circulations and North Pacific Intermediate Water (NPIW) in the North Pacific. The red thick dashed line indicates transformation of Okhotsk Sea Intermediate Water (OSIW) by cabbeling the subtropical NPIW along the subarctic-tropical frontal zone (You, 2003). The light brown solid line with arrow indicates the spreading path of subtropical NPIW from northeast North Pacific southward toward the low-latitude northwest North Pacific (You, 2003). Yellow solid lines with arrow represent two passages through which NPIW enter into the Okinawa Trough. This figure was created with Ocean Data View (odv.awi.de). (b) Location of sediment core CSH1 investigated in this study (red diamond). Also shown are locations of sediment cores investigated previously from the Okinawa Trough (PN-3,E017,255 and MD012404; white cross line), the northern and southern Japan (GH08-2004 and GH02-1030), the Bering Sea (PC-23A) and the northeastern Pacific (ODP167-1017). The detailed information for these cores can be seen in Table 1.

**Figure 2.** Spatial distribution of sea surface salinity in the East China Sea. (a) summer (July to September); (b) winter (January to March). Lower sea surface salinity in summer relative to that of winter indicates strong effects of summer East Asian Monsoon.





**Figure 3.** (a) Lithology and oxygen isotope ($\delta^{18}$O) profile of planktic foraminiferal species *Globigerinoides ruber* (*G.ruber*) in core CSH1. (b) Plot of ages versus depth for core CSH1. Three known ash layers are indicated by solid red rectangles. (c) Time series of linear sedimentation rate (LSR) from core CSH1. (d) Comparison of age model of core CSH1 with Chinese Stalagmite composite $\delta^{18}$O curve of (Cheng et al., 2016).Tie points for CSH1 core chronology (Table 2) in Figures2b and 2c are designated by blue and red solid dots.

**Figure 4.** Age versus (a) linear sedimentation rate (LSR), (b) C/N molar ratio, (c) Total organic carbon (TOC) concentration, (d) Total nitrogen (TN) concentration, (e) CaCO$_3$ concentration, (f) Al concentration, (g) Mn concentration, (h) Mo/Mn ratio, (i) Mo concentration, (j) excess Mo concentration, (k) U concentration and (l) excess U concentration in core CSH1. Gray and black vertical bars indicate different sediment intervals in core CSH1. MIS indicates Marine Isotope Stage. 8.2 ka, PB, YD, B/A, HS1, LGM and HS2 refer to 8,200 year cold event, Preboreal, Younger Dryas, Bölling - Alleröd, Heinrich Stadial 1, Last Glacial Maximum and Heinrich Stadial 2, respectively, which were identified in core CSH1. Blue solid diamonds in Figure 3l indicate the age control points.

**Figure 5.** Scatter plots of Mo$_{excess}$ vs Mn concentrations and U$_{excess}$ concentration vs Mo/Mn ratio at different time intervals in core CSH1. A various correlation is present in core CSH1 at different time intervals, which shows their complicated geochemical behaviors. Strong positive correlation between U$_{excess}$ concentration and Mo/Mn ratio suggest that they are suitable to track redox conditions in the past.

**Figure 6.** Proxy-related reconstructions of intermediate water oxygenation at site CSH1 (this study) compared with oxygenation records from other locations of the North Pacific and published climatic and environmental records from the Okinawa Trough. From top to bottom: (a) CaCO$_3$ concentration, (b) U$_{excess}$ concentration, (c) Mo/Mn ratio, (d) Mn concentration and (e) abundance of *P.obliquiloculata* in core





CSH1 (Shi et al., 2014) and (f) $\delta^{15}$N of TOC in core MD01-2404 (Kao et al., 2008), (g)
$\delta^{13}$C of *C.wuellerstorfi* in core PN-3(Wahyudi and Minagawa, 1997), (h) Dysoxic taxa
(%) in core ODP 167-1017 in the northeastern Pacific (Cannariato and Kennett, 1999)
and (i) $\delta^{13}$C of *Uvigerina akitaensisthe* in core PC23A in the Bering Sea (Rella et al.,
2012). Gray and black vertical bars are the same as those in Figure 4.

**Figure 7.** Proxy records favoring the existence of oceanic ventilation seesaw between
the subtropical North Pacific and North Atlantic during the last deglaciation and
enhanced carbon storage at mid-depth waters. (a) Atmospheric $CO_2$ concentration
(Marcott et al., 2014) (b) Indicator of strength of Atlantic Meridional Ocean
Circulation ($^{231}$Pa/$^{230}$Th) (Bohm et al., 2015; McManus et al., 2004); (c) benthic $\delta^{13}$C
record in core PC-23A in the Bering Sea (Rella et al., 2012); (d) Mo/Mn ratio in core
CSH1; (e) U $_{excess}$ concentration in core CH1. Blue diamonds are the same as those in
Figure 3.



Table 1

| Label in Figure 1b | Station | Latitude(°N) | Longitude(°E) | Water depth (m) | Area | Reference |
|---|---|---|---|---|---|---|
| | CSH1 | 31.23 | 128.72 | 703 | Okinawa Trough | this study |
| A | PN-3 | 28.10 | 127.34 | 1058 | Okinawa Trough | Wahyudi and Minagawa, (1997) |
| B | MD012404 | 26.65 | 125.81 | 1397 | Okinawa Trough | Kao et al., (2008) |
| C | E017 | 26.57 | 126.02 | 1826 | Okinawa Trough | Li et al., (2005) |
| D | 255 | 25.20 | 123.12 | 1575 | Okinawa Trough | Jian et al., (1996) |
| E | GH08-2004 | 26.21 | 127.09 | 1166 | East of Ryukyu Island | Kubota et al. (2010) |
| | GH02-1030 | 42.23 | 144.21 | 1212 | Off Japan | Sagawa and Ikehara, (2008) |
| | PC-23A | 60.16 | 179.46 | 1002 | Bering Sea | Rella et al.,(2012) |
| | ODP167-1017 | 34.54 | 239.11 | 955 | NE Pacific | Cannariato and Kennett, (1999) |





1  Table 2

| Depth(cm) | AMS $^{14}$C (yr) | Error (yr) | Calibrated Age (yr) | Tie Point Type | LSR (cm/ka) | Source |
|---|---|---|---|---|---|---|
| 10 | 3420 | ±35 | 3296 | $^{14}$C | | Shi et al., (2014) |
| 106 | 7060 | ± 40 | 7545 | $^{14}$C | 22.59 | Shi et al., (2014) |
| 218 | | | 12352 | Stalagmite, YD | 23.30 | This study |
| 322 | | | 16029 | Stalagmite, H1 | 28.28 | This study |
| 362 | | | 19838 | Stalagmite | 10.50 | This study |
| 466 | | | 23476 | Stalagmite, DO2 | 28.59 | This study |
| 506 | | | 24163 | Stalagmite, H2 | 58.22 | This study |
| 698 | | | 28963 | Stalagmite, DO4 | 40.00 | This study |
| 746 | | | 29995 | Stalagmite, H3 | 46.51 | This study |
| 834 | | | 32442 | Stalagmite, DO5 | 35.96 | This study |
| 938 | | | 37526 | Stalagmite, DO8 | 20.46 | This study |
| 978 | | | 39468 | Stalagmite, H4 | 20.60 | This study |
| 1058 | | | 46151 | Stalagmite, DO12 | 11.97 | This study |
| 1122 | | | 49432 | Stalagmite, DO13 | 19.51 | This study |
| 1242 | | | 52831 | Stalagmite, DO14 | 35.30 | This study |
| 1282 | | | 57241 | Stalagmite, DO16 | 9.07 | This study |
| 1346 | | | 61007 | Stalagmite, H6 | 16.99 | This study |
| 1530 | | ±2590 | 73910 | MIS4/5 | 14.26 | Shi et al., (2014) |
| 1610 | | ±3580 | 79250 | MIS 5.1 | 14.98 | Shi et al., (2014) |



5    Fig.1

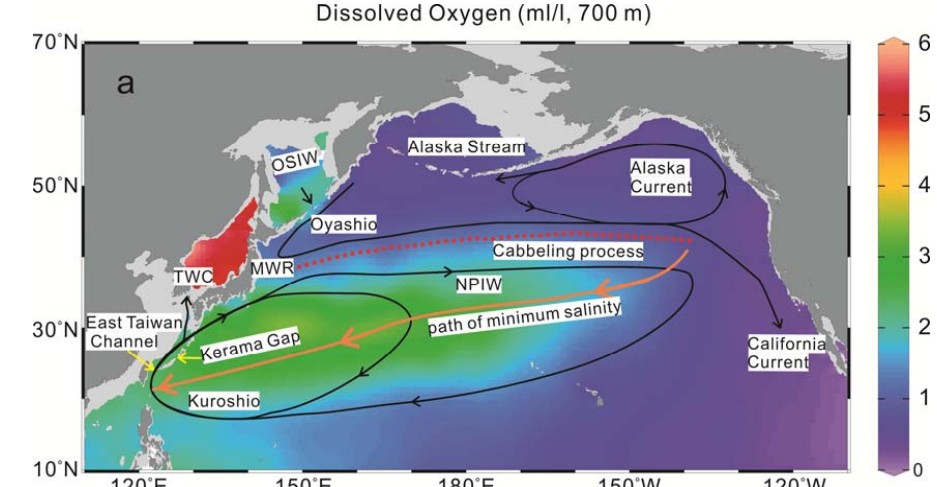

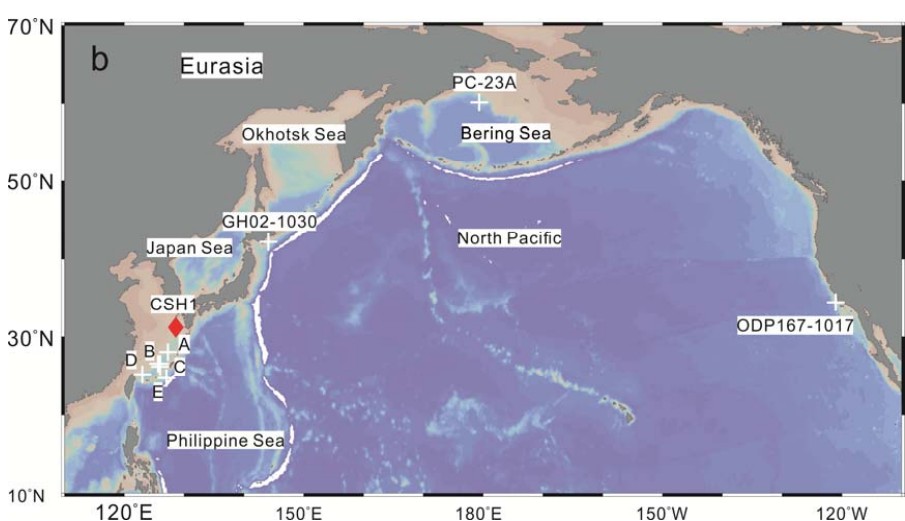



Fig.2

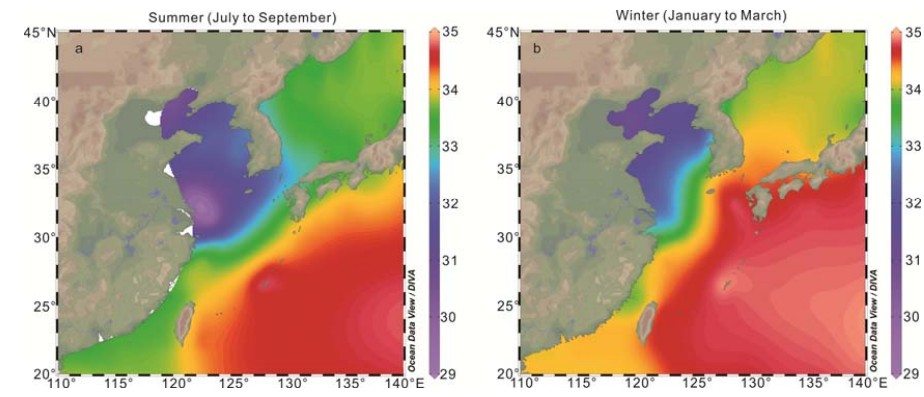

Fig.3

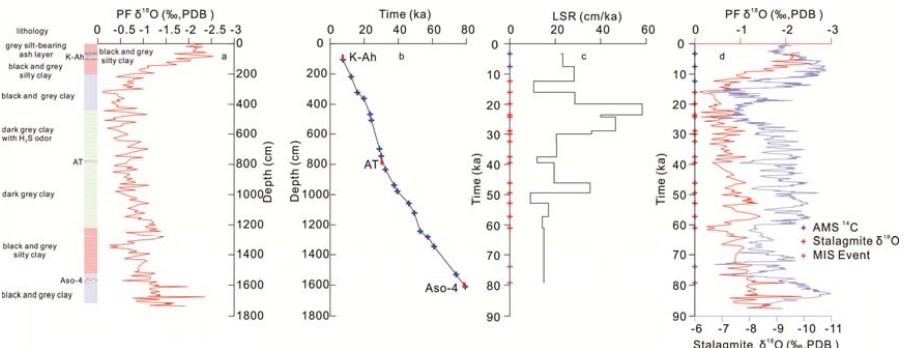





Fig.4

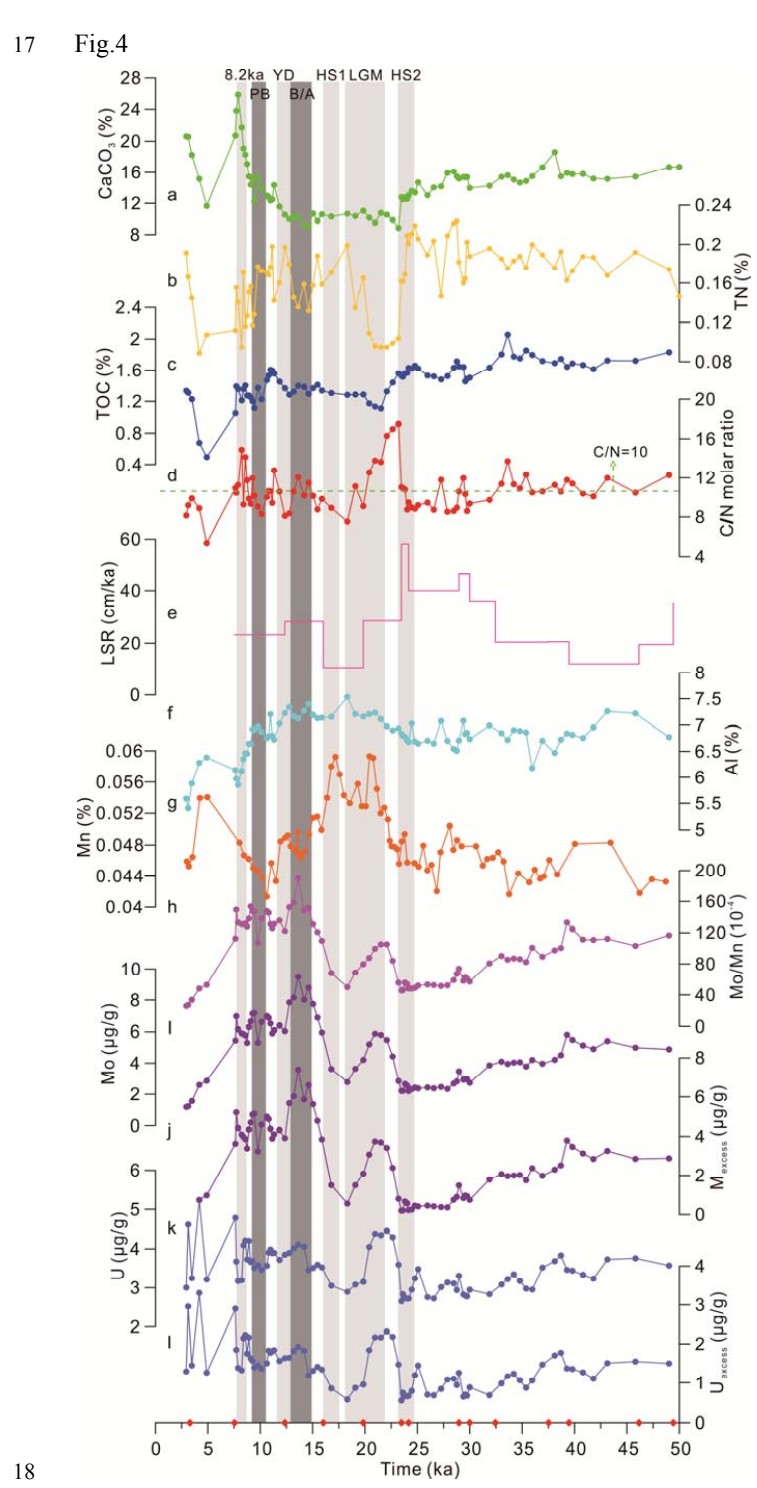



Fig.5

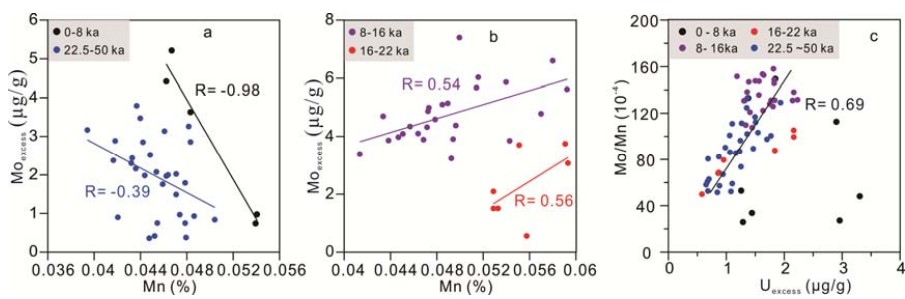





Fig.6

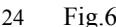



Fig.7

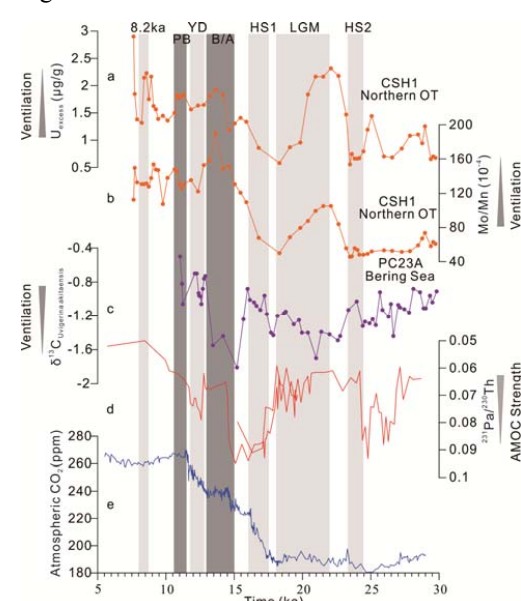

