# Peer review of "Millennial-scale variations of sedimentary oxygenation in the western"

_Climate of the Past, 2019_

## Referee Comment (RC1) · Anonymous Referee #1 · 18 Jul 2019

Review Zou et al 'Millennial-scale variations of sedimentary oxygenation in the western subtropical North Pacific and its link to North Atlantic climate.

Zou et al. study sedimentary redox conditions in the Okinawa Trough, and use this as a proxy to infer bottom water oxygen concentrations (I think, it is not always very clear from the manuscript). The title promises more than the paper delivers; the link with North Atlantic climate is only mentioned briefly and explanation is sometimes unclear.

The authors present interesting data, but the paper itself needs work. There is so much information (several times incorrectly referenced), and some information seems irrelevant. There is also a lot of internal discussion within the paper without reaching

firm conclusions. While the authors are critical about their own proxy, they are less so about others and this needs to be improved.

Comments: Authors should check that their references are appropriate. Several references are not put in the right context.

There are several other studies that deal with the North Pacific and NPIW, which are not referenced here; this includes work by Rippert et al. (2017), Max et al. (2017)

Abstract: Lines 44-50: these sentences go around the bushes. Really what you want to say is that sedimentary oxygenation conditions at mid-depth in the subtropical western North Pacific were more or less similar over the last 50,000 years, apart from the Bolling-Allerod and Pre-boreal. However, it may not be possible to compare with Holocene data, as this may be compromised by ash. This is not made very clear in the manuscript.

Lines 59-61: how does it seem to be driven? Is this not something you are proposing? Then it is not seem.

The authors mix up NADW and the Atlantic Meridional Overturning Circulation. For a good description of AMOC see recent paper by Frajke-Williams et al. (2019).

Introduction: Lines 70-73: not sure how to interpret this. Where is the respired carbon stored? At the sediment-seawater interface, in sedimentary pore-waters, or in seawater? The study of Lu et al. (2016) deals with I/Ca in planktonic foraminifer in the Pacific Sector of the Southern Ocean, to reconstruct upper ocean oxygenation, the part of respired carbon in their paper refers to a different study (Hoogakker et al., 2015).

Lines 76-83: the study of Cartapanis is from the northeastern Pacific, but not high latitude or subarctic.

Line 92: explain what cabbeling is, not everyone will have heard of the term. Lines 95-97: do the data really show this? The one core at $\sim$ 1km is about 0.04 per mil lighter (and within error), but crucially there is no Holocene equivalent for the 0.7 km core.

Line2 149-152: do you mean 'is governed by' instead of 'is the balance between'.

Figure 1. O2 map, and locations of cores. Are all the cores discussed in the paper? Is it worrying that the main core CSH1 is from just south of Japan and perhaps should not be considered an open ocean core? What do the letters A to E stand for?

Setting: Do details of discharge and SSS add anything to this study?

Material and methods: What causes the high accumulation rates in this core? As the accumulation rates vary significantly, how do these influence the patterns in redox elements etc?

Lines 339-340: preservation of TOC and CaCO3 are influenced by many factors and not a widely used paleo-export proxy.

Line 354-357: have you checked that it is an extant biological component that makes up the high CaCO3 going from B/A to $\sim$ 8000 years? Can you explain the differences in the LSR figures between Figures 3 and 4? For examples, in Figure 3 highest rates occur centred around 22 kyrs as part of a large interval of high LSR (from 30 to 20), whereas in Figure 4 this occurs earlier (33 to 24 kyrs).

Lines 365-266: if U concentrations are affected by volcanic material over the last 8.5 yrs, then surely so are other sedimentary properties? I would like to see an argument in the main text discussing why certain proxy methods are deemed not to be influenced by this volcanic material, whilst others are. If it turns out that interval should not be used than this creates the complication of not being able to compare the down core data with more modern.

Line 370: change 'seems' to 'may'.

Lines 372-377: are there any other studies that use Mo/Mn ratios as a sedimentary oxygenation proxy, to support your interpretation?

Line 380: define oxygen deficient.

[Figure]

Discussion: Lines 387-392: I would recommend the authors to use more appropriate scheme that is used for sea-water that includes hypoxic, for example as defined by Bianchi et al. (2012). You will also found that suboxic is classified as < 2-10 $\mu$mol/l.

Lines 403-405: I do not understand this reasoning. You have not linked weakly restricted basin settings with euxinia? It is confusing talking about ppm in the main text, whilst Figure 4 gives concentrations in $\mu$g/g.

Lines 406-412: Mo/U ratios are not shown in the manuscript. This is out of the blue.

Lines 413-425: two studies. More importantly though, Figure 4 shows no benthic foraminifera data, and it is therefore impossible to confirm this claim of ventilation pattern from benthic foraminiferal assemblages to be similar to that of the RSEs. It would also be good to see a more critical discussion about this proxy.

Lines 425-428: No. There is at least 800 meter water depth difference between your core and the others. The core of the current study is situated just above the low oxygen zone, whereas those of the other two studies are in /below the low oxygen zone.

Lines 439-448: why are you looking at one NE Pacific to find out what is happening at your core site? There are several studies from across the Pacific that show something happening around the same time period (for example see Moffit et al., 2015), and Galbriath and Jaccard (2015), so rather than repeating the same discussion for a very small area, it would be easier to build up on those results.

Lines 454-457: no, at those high temperatures you would only get a reduction in O2 of $\sim$ 3 for one degree warming, and 15 for a four degree warming (assuming no large salinity changes). Higher glacial salinity would cause less reduction in O2.

Lines 457-458: sentence does not make sense.

Lines 848-846: does not make sense. How does subsurface water oxygen consumption lead to lower oxygen concentrations in deeper waters?

Lines 491-494: again not taking into account other factors that influence CaCO3 accumulation and preservation in sediments. For discussion on Kuroshio Current: see Lim et al. (2017).

Line 544-550: coined? Matsumoto et al. (2002) discuss one radiocarbon age from the Santa Barbara basin in relation to oxygen content, but at no point do they propose that GNPIW was stronger oxygenated. Cartapanis et al. (2011) and Ohkushi et al. (2013) discuss that the NE OMZ Pacific strengthened and weakened at millennial time scales, not glacial interglacial timescales. Also around the equatorial Pacific it is suggested that there was no difference in intermediate water oxygenation between the last glacial and Holocene (Hoogakker et al., 2018). Further down in the South Pacific Lu et al. (2016) suggest that upper waters were depleted in oxygen during the last glacial.

Lines 550-556: generalised comment, what about brine (aka Kim et al. 2011).

Lines 556-559: what is intensified GNPIW?

Discussion lines 560-571: needs tightening, it is unclear where this goes and how it relates to this study?

Line 629: what is ocean ventilation seesaw? There is hardly any explanation for this in the main text, and Figure 7 shows strength of AMOC in the Atlantic and compares this with the current study.

---

## Referee Comment (RC2) · Anonymous Referee #2 · 24 Jul 2019

Zou et al., present a rather interesting study focusing on reconstructing the oxygenation history in the Okinawa Trough covering the last 50 kyrs. Specifically, the authors attempt to disentangle the typically confounding influence of export production (and by inference the oxygen consumption related to organic matter degradation) and bottom water ventilation on the sedimentary redox condition. Their geochemical records largely corroborate previous findings in that oxygenation patterns at intermediate depths in the North Pacific were primarily controlled by the production and ventilation of North Pacific Intermediate Water. Specifically, conditions were generally better oxygenated during stadials, when NPIW was generally better ventilated and vertically expanded. Furthermore, their data support a general expansion of oxygen-depleted waters at in-

termediate depths during the B/A occupying large swaths of the North Pacific.

The manuscript is well documented and quite detailed in places. The argumentation could be somewhat streamlined (see comments below) and would certainly benefit from editorial support. I would also recommend the argumentation to focus on the aspects outlined in the title and abstract.

l. 35 – deep ocean carbon sequestration is certainly one the reasons potentially explaining lower glacial pCO2 concentrations, but certainly not the only one. Please rephrase to avoid unnecessary confusion

l. 36 – I would suggest rephrasing as follows – However, the potential role of subtropical North Pacific subsurface waters in modulating. . .

l. 48 and throughout – why is HS1 so much different from HS2 when considering their respective oxygenation history?

l. 62 – agreed. But I would add that in addition of the flushing of a poorly ventilated deep water mass upon the resumption of NADW, many export production records show a drastic increase during the B/A (e.g. Kohfeld & Chase, 2011), which could account for enhance oxygen removal associated with organic matter respiration upstream of the core site location.

l. 70 – AT the sediment-water interface

l. 83 . . . in marine sediment cores

l. 86 . . . in the subarctic North Pacific.

l. 92 – I would suggest to briefly explain what cabbeling means

l. 96 and throughout (incl. Fig. 6) – this may well be a semantic issue, but benthic d13C cannot be considered as a ventilation proxy per se, as the isotopic value can be obfuscated by air-sea gas exchange in locations where subsurface water masses form.

l. 151 – processes that are related to the supply of oxygen by ocean circulation and. . .

l. 163 – technically one should specify that under reducing conditions it is the authigenic fraction of Mn (as opposed to its detrital background) that remains low.

l. 168 – please add adequate reference

l. 195 – volcanism

§3 – I would suggest to substantially shorten this paragraph as the general oceanographic setting is already outlined in the introduction. I would recommend focusing on the aspects directly relevant to the argumentation (nutrient, dissolved O2).

l. 287 – why is the sedimentation rate so high during HS2 when both export production and detrital input (based on Al) are low? I would suggest verifying the age pointers for that interval.

l.291 – 85 samples covering 50 kyrs cannot provide an average time resolution of 200 yrs.

l. 345-346 – Maybe. But it may also suggest that the sedimentary CaCO3 content could be directly controlled by dilution. I would interpret the export productivity records with caution.

l.371-372 – interesting idea!

l. 427 – supply seems more adequate than provision.

l. 487-488 – please also consider citing Galbraith et al., 2007.

l. 489-490 – please keep in mind that O2 can be consumed upstream of the core site location as the removal of O2 in relation to organic matter degradation is integrated over the flow path of a give subsurface water mass.

Fig. 6C – shouldn't the grey triangle right of the vertical axis be flipped upside down (i.e. high Mo/Mn coincident with low oxygenation)?

[Figure]

---

## Author Comment (AC1) · 26 Aug 2019

Anonymous Referee #1 Review Zou et al 'Millennial-scale variations of sedimentary oxygenation in the westernsubtropical North Pacific and its link to North Atlantic climate.

Zou et al. study sedimentary redox conditions in the Okinawa Trough, and use this as a proxy to infer bottom water oxygen concentrations (I think, it is not always very clear from the manuscript). The title promises more than the paper delivers; the link with North Atlantic climate is only mentioned briefly and explanation is sometimes unclear. The authors present interesting data, but the paper itself needs work. There is so

much information (several times incorrectly referenced), and some information seems irrelevant. There is also a lot of internal discussion within the paper without reaching firm conclusions. While the authors are critical about their own proxy, they are less so about others and this needs to be improved.

Reply#1: We thank Reviewer#1 for the time to review our manuscript with constructive comments, which contribute improving our manuscript. Our work adds an important element to the process in the subtropical North Pacific during the last deglaciation. Our data suggest a substantial impact of NPIW on sedimentary oxygenation in the subtropical North Pacific and also an expansion of oxygen minimum zone in the western North Pacific during the B/A. According to the comments of Reviewers #1 and #2, we have made a thorough revision on our manuscript from the abstract to the conclusions, including Figures 3-7, and use "track changes" to display our revised text in red color (supplement 1). In the revised manuscript (supplement 2), we reworded the abstract, amended the age model (see new Figure 3), rephrased some sentences and paragraphs, and added new evidence of benthic foraminifera abundance of core E017 (see new Figure 6) and added some discussion on the North Atlantic Climate. We hope that the Reviewer will find the new finding clearer. In the following we provide point-by-point response to all the Reviewer's comments as well as excerpts from the manuscript.

Comments: Authors should check that their references are appropriate. Several references are not put in the right context.

Reply#2: Thanks. Checked and Revised. The following references were included in the revised MS: in Line 65 for reference Hoogakker et al., 2015 in Line 74 for reference Addison et al., 2012 in Line 97 for reference Max et al 2017 in Line 100 for reference Rippert et al. 2017 in Line 109 for reference Sibuet et al., 1987 in Line 171 for reference Morford and Emerson, 1999 in Line 336 for references Cartapanis et al., 2011; Lembke-Jene et al., 2017 in Line 351 for reference Shi et al., 2014 in Line 359 for reference Chang et al., 2009 in Line 371 for reference Zhu et al., 2015 in Line 376 for reference Lim et al., 2017 in line 400 for reference Bianchi et al. 2012 in lines 450-451

for references Galbraith and Jaccard, 2015; Moffitt et al., 2015; Praetorius et al., 2015 in line 471 for reference Brewer and Peltzer, 2016 in line 502 for reference Galbraith et al., 2007 in line 536 for reference Lim et al., 2017 in line 541 for reference Andres et al., 2015 in line 557 for reference Kubota et al., 2015 in line 584 for reference Kubota et al., 2015 in line 629 for reference Lynch-Stieglitz, 2017 in lines 631-635 for references Böhm et al., 2015; McManus et al., 2004; Liu et al., 2009; Zhang et al., 2017; Barker et al., 2010; Knorr and Lohmann, 2007; in lines 648 for reference Lohmann et al., 2019 in lines 705-706 for references Galbraith and Jaccard, 2015; Jaccard and Galbraith, 2012; Moffitt et al., 2015

The following original references were removed in the revised MS: in line 65 Lu et al., 2016; in line 400 Savrda and Bottjer, 1991 in line 471 Benson and Krause, 1984 in line 541 Matsumotoet al. (2002) in line 557 Kubota et al., 2010 in line 565 Wang and Wang, 2008 in line 584 Kubota et al., 2010

There are several other studies that deal with the North Pacific and NPIW, which are not referenced here; this includes work by Rippert et al. (2017), Max et al. (2017)

Reply#3: Thanks. These two references have been included in the revised manuscript and compared with their findings. In particular, both two papers highlight the substantial effects of NPIW on subsurface water composition of Eastern Tropical North Pacific and potential roles in regulating global climate. Likewise, the effect of NPIW's ventilation on the western subtropical North Pacific in our study is observed clearly. Our study further validates the role of NPIW in determining its downstream ocean environment.

We have added Lines 97-101 "More recently, Max et al (2017) identified the substantial effects of intensified NPIW on $\delta 13C$ of deep-dwelling planktic foraminifera Globorotaloideshexagonusin the Eastern Equatorial Pacific during Marine Isotope Stage (MIS) 2. Subsequently, Rippert et al. (2017) confirmed that such enhanced effect of NPIW also occurred during MIS 6. "

Abstract: Lines 44-50: these sentences go around the bushes. Really what you want

to say is that sedimentary oxygenation conditions at mid-depth in the subtropical western North Pacific were more or less similar over the last 50,000 years, apart from the Bolling-Allerod and Pre-boreal. However, it may not be possible to compare with Holocene data, as this may be compromised by ash. This is not made very clear in the manuscript.

Reply#4: Thanks for your suggestion. We have rephrased these sentences in the revised manuscript. The sentence was amended as follows. The text was modified as follows. Lines 42-45: "Our results suggest that enhanced sedimentary oxygenation at mid-depths of the subtropical North Pacific occurred during the cold intervals and after 8.5 ka, while decreased oxygenation during the Bölling-Alleröd (B/A) and Preboreal."

As suggested by the Reviewer, our data suggest well-oxygenated water during cold intervals apart from the B/A and Preboreal. For the Holocene, increased sedimentary oxygenation is ascribed to an intensified Kuroshio system, although discrete volcanic materials, indicating by positive Eu anomaly (Zhu et al., 2015) and more radiogenic Nd isotopes (unpublished data), would dilute the Holocene data. Modern observations suggest the Kuroshio can reach the seafloor at 1200 m in the East China Sea (Andres et al., 2015). In the geological past, both various proxy data and modeling simulations suggest an intensified Kuroshio reentering the OT at early Holocene 9-9.6 ka (Chang et al., 2015; Diekmann et al., 2008; Dou et al., 2016; Lim et al., 2017; Zheng et al., 2016b). In our previous study based on the same core CSH1 (Zhu et al., 2015), we have suggested that the discrete volcanic materials during the Holocene is closely related to enhanced Kuroshio intensity events. Recently, increased total Hg concentration in the sediments from the middle Okinawa since 9.3 ka (Lim et al., 2017) was also suggested and explained by hydrothermal Hg source (due to much higher concentrations than potential terrigenous end-members) brought to the site location via intensified Kuroshio Current. Therefore, Kuroshio Current is the fundamental mechanism to improve the ventilation of the OT, not the effects of volcanic material dilution.

We have now added more information into the section 5.2 Redox-sensitive Elements.

Lines 369-376: "It should be noted that pronounced variations in U concentration since 8.5 ka is related to the occurrence of discrete volcanic materials.A significant positive Eu anomaly (Zhu et al., 2015) together with more radiogenic Nd values (unpublished data) from the same core confirms the occurrence of discrete volcanic materials and its dilution effects on terrigenous components since 7 ka. Occurrence of discrete volcanic material is likely related to intensified Kuroshio Current during the mid-late Holocene, as supported by higher hydrothermal Hg concentrations in sediments from the middle OT (Lim et al., 2017)."

Lines 59-61: how does it seem to be driven? Is this not something you are proposing? Then it is not seem.

Reply#5: Thanks. Revised."seem to be driven" is replaced by "can be driven". Lines 49-51 in the revised MS. The sentence now reads: "The enhanced formation of NPIW during HS1 can be driven by the perturbation of sea ice formation and sea surface salinity oscillation in high-latitude North Pacific."

The authors mix up NADW and the Atlantic Meridional Overturning Circulation. For a good description of AMOC see recent paper by Frajke-Williams et al. (2019).

Reply#6: Thanks. We checked the paper Frajke-Williams et al. (2019). In the revised manuscript, we replaced AMOC by North Atlantic Deep Water and reworded these sentences. Lines 51-57 in the revised MS. The sentences now read: "The diminished sedimentary oxygenation during the B/A due to upwelling of aged, nutrient-rich deep water and enhanced export production, indicates an enhanced $CO_2$ sequestration at mid-depth waters, along with a slight increase in atmospheric $CO_2$ concentration. We attribute these millennial-scale changes to intensified NPIW and enhanced abyss flushing during deglacial cold and warm intervals, respectively, on the basis of background climate change due to shift in North Atlantic Deep Water formation."

Introduction: Lines 70-73: not sure how to interpret this. Where is the respired carbon stored? At the sediment-seawater interface, in sedimentary pore-waters, or in seawater? The study of Lu et al. (2016) deals with I/Ca in planktic foraminifer in the Pacific Sector of the Southern Ocean, to reconstruct upper ocean oxygenation, the part of respired carbon in their paper refers to a different study (Hoogakker et al., 2015).

Reply#7: We have reworded the first sentence for clearer statement. Additionally, the reference (Lu et al., 2016) was replaced by reference (Hoogakker et al., 2015).

The sentence was amended as follows, Lines 62-66: "The sluggish ocean ventilation and efficient biological pump in the ocean facilitate carbon sequestration in the ocean interior, linking to atmospheric $CO_2$ drawdown, which in turn play a crucial role in regulating sedimentary oxygen on millennial and orbital timescales (Hoogakker et al., 2015; Jaccard and Galbraith, 2012; Sigman and Boyle, 2000)."

Lines 76-83: the study of Cartapanis is from the northeastern Pacific, but not high latitude or subarctic.

Reply#8: The study by Cartapanis et al. (2011) presents high-resolution redox-sensitive trace metals in sediment core from the Eastern Tropical North Pacific and found improved intermediate water oxygenation during Heinrich events. The enhanced ventilation during Heinrich intervals is consistent with our inference. Therefore, we would like to keep this reference in the revised manuscript and rephrased the expression of the context.

Now the sentence reads: "Previous studies from eastern and western North Pacific margins and subarctic Pacific have identified drastic variations in export productivity and ocean oxygen levels at glacial-interglacial timescales using diverse proxies such as trace elements (Cartapanis et al., 2011; Chang et al., 2014; Jaccard et al., 2009; Zou et al., 2012), ......"

Line 92: explain what cabbeling is, not everyone will have heard of the term.

Reply#9: Done. Cabbeling is a mixing process to form a new water mass with increased density than that of parent water masses. We've provided a definition see

Lines 85-86 in the revised MS.

The sentence was changed to: "...... that cabbeling, a mixing process to form a new water mass with increased density than that of parent water masses, is the principle mechanism responsible for ......"

Lines 95- 97: do the data really show this? The one core at 1km is about 0.04 per mil lighter (and within error), but crucially there is no Holocene equivalent for the 0.7 km core.

Reply#10:These two studies show enhanced formation of North Pacific Intermediate Water during the last glaciation. On the basis of >30 sediment cores on the northern Emperor Seamounts and in the Okhotsk Sea with a water depth of 1000 to 4000 m, Keigwin (1998) found that there was a better ventilated water mass above 2000 m in the northwestern Pacific and it was characterized by relatively fresher compared to deep waters. Matsumoto et al. (2002) examined nutrient proxies in sediment cores with water depth of 740m to 3320 m in the North Pacific and found a presence of distinctive water masses below and above 2000m water depth in the glacial Pacific with higher benthic $\delta$13C in the upper 2000m water.

Line2 149-152: do you mean 'is governed by' instead of 'is the balance between'.

Reply#11: Revised. Lines 151-154 in the revised MS.

Now the sentence reads: "Sedimentary redox condition is governed by the rate of oxygen supply from the overlying bottom water and the rate of oxygen removal from pore water (Jaccard et al., 2016), processes that are related to the supply of oxygen by ocean circulation and organic matter respiration, respectively."

Figure 1. O2 map, and locations of cores. Are all the cores discussed in the paper? Is it worrying that the main core CSH1 is from just south of Japan and perhaps should not be considered an open ocean core? What do the letters A to E stand for?

Reply#12: All cores shown in Figure1b have been discussed in this manuscript. For

example, benthic $\delta$13C in core PN-3 was used to indicate ventilation change in the OT. Concentrations of CaCO3 and reactive phosphorus recorded in core MD01-2404 were used to correlate with our productivity proxy. Benthic foraminiferal assemblages in cores E017 and 225 retrieved from the middle and southern OT were used to indicate the ventilation of deep water in the OT. Deep-water temperatures in cores GH08-2004 and GH02-1030, epibenthic $\delta$13C in core PC23A and benthic foraminiferal assemblages in ODP 1617-1017 have implicated the ventilation of North Pacific Intermediate Water.

Core CSH1 is situated in the northern Okinawa Trough at depth of 703 m. In this area, both surface and deep water can be continuously replenished by water mass from open oceans.

Letters A to E stand for sediment cores previously reported in and near the Okinawa Trough and are shown in Table 1.

Setting: Do details of discharge and SSS add anything to this study?

Reply#13: Sea surface salinity (SSS) in the East China Sea is closely related to the precipitation intensity controlled by summer East Asian Monsoon (EAM). Accordingly, a recent study from the northern Okinawa Trough (U1429) has extended this relationship back to 400 ka(Clemens et al., 2018). In order to discern millennial-scale variability, we correlate planktic $\delta$18O of core CSH1 with Chinese stalagmite $\delta$18O, an indicator of summer EAM to establish the age model for core CSH1 in our study. We would like to add the details of discharge and SSS in the manuscript to let the readers know the details.

Material and methods: What causes the high accumulation rates in this core? As the accumulation rates vary significantly, how do these influence the patterns in redox elements etc?

Reply#14: In the northern Okinawa Trough, previous sediment provenance studies suggested the terrigenous sediments are mainly sourced from the Yellow and Changjiang Rivers, from China and short mountainous rivers (Japanese or even Taiwanese rivers) from the surrounding islands, as well as eolian dust, volcanic and hydrothermal materials from Yellow River, Changjiang and part of Korea and Japan Rivers (Beny et al., 2018; Li et al., 2015; Zhao et al., 2018; Zhao et al., 2017). Variation in sedimentation rate has been attributed to changes in eustatic sea level, EAM intensity, path and/or intensity of Kuroshio Current. Generally, sea level is thought to be the first-order factor for controlling linear sedimentation rate changes (Beny et al., 2018; Li et al., 2015; Zhao et al., 2017).

Our data show that there is no coherent relationship between linear sedimentation rate and concentrations of redox sensitive elements. For example, high sedimentation rate between 24.2 ka and 32.4 ka (around 40 cm/ka) corresponds to decreasing concentrations of redox sensitive elements. On the other hand, lower sedimentation between 16 ka and 19.8 ka also corresponds to lower concentrations of redox sensitive elements. Therefore, linear sedimentation rate is not a crucial factor in controlling concentrations of redox sensitive elements in core CSH1.

Lines 339-340: preservation of TOC and CaCO3 are influenced by many factors and not a widely used paleo-export proxy.

Reply#15: We fully agree that the preservation of TOC and CaCO3 are influenced by many factors, including supply, dissolution, organic matter degradation, terrigenous dilution, etc. Some factors can be ruled out and at times these two proxies have been used to reconstruct export productivity. In this study, C/N molar shows substantial contribution of terrigenous organic matter to total organic carbon, so it is not a suitable proxy for productivity reconstruction.

In the revised manuscript, we showed multiple lines of evidence to support the utility of CaCO3 as a reliable productivity proxy, including (1) strong negative correlation with terrigenous Al of core CSH1; (2) weak dilution effect of terrigenous material on

[Figure]

CaCO3; (3) similar pattern to sea surface temperature of core CSH1 (Shi et al., 2014) (the data have been included in Figure 6); (4) similar deglacial trends in CaCO3 and reactive phosphorus reported in core MD012404 retrieved from the middle OT (Chang et al., 2009; Li et al., 2017). All these lines of evidence support CaCO3 as a proxy for productivity.

We have added these information in lines 342-361 in the revised MS: "Several lines of evidence support CaCO3 as a reliable productivity proxy, particularly during the last deglaciation. The strong negative correlation coefficient (r = − 0.85, p<0.01) between Al and CaCO3 in sediments throughout core CSH1 confirms the biogenic origin of CaCO3 against terrigenous Al (Figure 4f). Generally, terrigenous dilution decreases the concentrations of CaCO3. Inconsistent relationship between percentage CaCO3 and sedimentation rate indicates a minor effect of dilution on CaCO3. Furthermore, the increasing trend in CaCO3 associated with high sedimentation rate during the last deglacial interval indicates a substantial increase in export productivity (Figures 4a and 4d). The high coherence between percent CaCO3 and alkenone-derived sea surface water (SST) (Shi et al., 2014) indicates a direct control on CaCO3 by SST. Moreover, a detailed comparison between CaCO3 concentrations and the previously published foraminiferal fragmentation ratio (Wu et al., 2004) shows, apart from a small portion within the LGM, no clear co-variation between them. These evidence suggest that CaCO3 changes are driven primarily by variations in carbonate primary production, and not overprinted by secondary processes, such as carbonate dissolution through changes in the lysocline depth and dilution by terrigenous materials. Likewise, similar deglacial trend in CaCO3 is also observed in core MD01-2404 (Chang et al., 2009), indicating a ubiquitous, not local picture in the OT. All these lines of evidence thus support CaCO3 of core CSH1 as a reliable productivity proxy to a first order approximation."

Line 354-357: have you checked that it is an extant biological component that makes up the high CaCO3 going from B/A to ∼ 8000 years? Can you explain the differences in the LSR figures between Figures 3 and 4? For examples, in Figure 3 highest rates

occur centred around 22 kyrs as part of a large interval of high LSR (from 30 to 20), whereas in Figure 4 this occurs earlier (33 to 24 kyrs).

Reply#16: Seven samples at 8.23 ka (120-124 cm), 9.26 ka (144-148 cm), 10.98 ka (184-188 cm), 11.66 ka (200-204 cm), 12.92 ka (232-236 cm), 14.05 ka (264-268 cm), and 15.18 ka (296-300 cm) in core CSH1 were analyzed by Modular Stereo Microscope (Zeiss SteREO Discovery V12) to look into the sediment components. It is clear that abundant biogenic tests, especially foraminiferal tests, are observed during these sediment intervals (Figure A1). On the other hand, increased concentration of CaCO3 is highly coherent with the abundance of planktic foraminifera species G.ruber and SST (Shi et al., 2014), indicating a substantial effect of SST on CaCO3.

In the original manuscript, we made a mistake for LSR in Figure 3. In the revised manuscript, this issue has been corrected (Figure 3).

In addition, high sedimentation (>40-60 cm/ka) in the original manuscript mainly occurred during HS2 and HS3. This can be caused by uncertainties of age control points at 23.476 ka (DO2) and 29.995 ka (H3). In the revised manuscript, these two age control points have been eliminated from the Chinese Stalagmite tuned age model. Even with this more conservative tuning approach, the conclusions on sedimentary oxygenation variations remain the same as before and robust.

Lines 365-266: if U concentrations are affected by volcanic material over the last 8.5 yrs, then surely so are other sedimentary properties? I would like to see an argument in the main text discussing why certain proxy methods are deemed not to be influenced by this volcanic material, whilst others are. If it turns out that interval should not be used than this creates the complication of not being able to compare the down core data with more modern.

Reply#17: The occurrence of volcanic material has been confirmed by positive Eu anomalies and more radiogenic $\varepsilon$Nd values (unpublished data) and it has substantial effects on concentration of terrigenous materials (Zhu et al., 2015). This argument has

been included in the revised text. Please see Reply#4.

Line 370: change 'seems' to 'may'.

Reply#18: Revised. "seem to" was replaced by "may". Line 380.

Lines 372-377: are there any other studies that use Mo/Mn ratios as a sedimentary oxygenation proxy, to support your interpretation?

Reply#19: To our knowledge, Mo/Mn ratios are not used as sedimentary oxygenation proxy in previous studies. In this study, we use both Mo/U ratio and excess U concentration to reconstruct sedimentary oxygenation changes. Among these, excess U concentration has been widely used for past sedimentary oxygenation changes (Jaccard and Galbraith, 2013; Jaccard et al., 2016; Jaccard et al., 2009). The strong positive correlation coefficient between excess U and Mo/Mn ratio in core CSH1 indicates its reliability and supports our interpretation. In addition, we also examined the Mo/Mn ratio in sediment cores from the Sea of Japan (KCES1, unpublished data), subarctic Pacific (LV76-18, unpublished data), and the Norwegian Sea (BB03, unpublished data). These data lend strong support for us to view the Mo/Mn ratio as a reliable proxy in reconstructing sedimentary oxygenation changes.

Line 380: define oxygen deficient. Reply#20:In the revised manuscript, "oxygen-deficient" was replaced by "deoxygenated".

The sentence now reads as follows, Lines 391: "...... together with lower Mn concentrations suggest a deoxygenated depositional condition during the late deglacial period (15.8 kaâ­Š9.5 ka)......" The oxygen thresholds are given in lines 399-401.

Discussion: Lines 387-392: I would recommend the authors to use more appropriate scheme that is used for sea-water that includes hypoxic, for example as defined by Bianchi et al. (2012). You will also found that suboxic is classified as < 2-10 $\mu$mol/l.

Reply#21: Thanks. The oxygen content scheme for seawater developed by Bianchi et al.(2012) has been adopted in the revised manuscript. Lines 400-402 in the revised

[Figure]

MS.

Following the Reviewer's suggestion we changed the sentence to: "Here, we adopt the definition of oxygen thresholds by Bianchi et al. (2012) for oxic (>120 $\mu$mol/kg O2), hypoxic (<60–120 $\mu$mol/kg O2) and suboxic (<2–10 $\mu$mol/kg O2) conditions, whereas anoxia is the absence of measurable oxygen."

Lines 403-405: I do not understand this reasoning. You have not linked weakly restricted basin settings with euxinia?

Reply#22: We thank the reviewer for this suggestion. The previous manuscript had an inaccurate statement. Now, we changed this sentence to make it more explicit. The sentence was amended as follows. Lines 413-415 in the revised MS. "Given that the northern Okinawa Trough is located in the open-ocean settings, we use the above mentioned two proxies to evaluate the degree of oxygenation in sediments."

It is confusing talking about ppm in the main text, whilst Figure 4 gives concentrations in $\mu$g/g.

Reply#23: Corrected.

In the revised manuscript, the uniform concentration unit, $\mu$g/g has been used for all trace elemental concentrations in the main text. Lines 416 and 421 in the revised MS.

Lines 406-412: Mo/U ratios are not shown in the manuscript. This is out of the blue.

Reply#24: Revised.

Mo/U ratio has been included in Figure 4m.

Lines 413-425: two studies. More importantly though, Figure 4 shows no benthic foraminifera data, and it is therefore impossible to confirm this claim of ventilation pattern from benthic foraminiferal assemblages to be similar to that of the RSEs. It would also be good to see a more critical discussion about this proxy.

Reply#25: We understand the point raised by the Reviewer. In this study, the benthic foraminiferal species are not counted in core CSH1. The benthic foraminiferal census data in cores E017 (1826 m water depth), 255 (1575 m water depth) are used to indicate the variations in ventilation of the middle and southern OT. The age model for core 255 (core length 655cm) was determined by two AMS 14C dates of N.dutertrei at depth 370 cm (9.17 ka) and 590 cm (18.8 ka) (Jian et al., 1996), whereas the age model for core E017 was established by 6 age points (Li et al., 2005). Although the downcore abundance of hypoxia-like species in both studies are similar to each other(Li et al., 2005), here we focus on the benthic foraminiferal census data in core E017. We recalibrated the AMS 14C dates using CALIB 7.04 software with Marine 13 calibration dataset (Reimer et al., 2013) and compare the profiles of oxygen-like species and hypoxia-like species with our Mo/Mn and excess U. For the sake of simplicity, the abundance of Bulimina aculeata (hypoxia-like species) and Cibicidoides hyalina (oxygen-like species) have been included in Figure 6 in the revised manuscript. High relative abundance of B.aculeata and low C. hyalina suggest the dominance of hypoxic environment, whereas the oxic condition is prevailed after ∼7 ka. This is consistent with our RSEs data, suggesting a widespread occurrence of oxygen-depleted water in the Okinawa Trough during the last deglaciation.

Lines 425-428: No. There is at least 800 meter water depth difference between your core and the others. The core of the current study is situated just above the low oxygen zone, whereas those of the other two studies are in /below the low oxygen zone.

Reply#26: Thanks for your comments. The seafloor bathymetry is much deeper in the southern OT, and shoals gradually toward the northern OT. Although our core is above the sill depth (1100m), while others below (1100m), previous investigations show the hydrographic characteristics in the mid-depth and deep OT are mainly regulated by the NPIW in the western boundary region of the Philippine Sea that flows into the OT through the Kerama Gap (1100 m) and the channel east of Taiwan (775 m) (Nakamura et al., 2013). Thus, ventilation signals recorded in these cores are mainly controlled by

the same physical processes, though export productivity in different areas also exerts some impacts on deepwaters and sedimentary oxygenation.

Lines 439-448: why are you looking at one NE Pacific to find out what is happening at your core site? There are several studies from across the Pacific that show something happening around the same time period (for example see Moffit et al., 2015), and Galbriath and Jaccard (2015), so rather than repeating the same discussion for a very small area, it would be easier to build up on those results.

Reply#27: At present, a contrasting distribution of dissolved oxygen concentration of the subsurface water can be observed in the eastern and western North Pacific margins (Figure 1), which is characterized by strong Oxygen Minimum Zone in the eastern margin and oxic condition in the western margin. The benthic foraminiferal assemblages from ODP site 1017 exhibit strengthening of OMZ during warm periods and weakening during cold periods (Cannariato and Kennett, 1999). The question that whether expanded OMZ can extend toward the western NW Pacific remains elusive in the geological past during warming intervals. In fact, the key question involved is how to explain the cause of oxygenation variation on basin-wide scale. Comparison of our results from the western North Pacific with those of the eastern North Pacific aids to understand the mechanism behind sedimentary oxygen changes.

We have added more information into the section 6.2 and all these references have been included in Lines 448-463. "These processes have been invoked in previous studies to explain the deglacial Pacific-wide variations in oxygenation by either one or a combination of these factors (Galbraith and Jaccard, 2015; Moffitt et al., 2015; Praetorius et al., 2015). Our data also suggest drastic variations in sedimentary oxygenation over the last 50 ka. However, the mechanisms responsible for sedimentary oxygenation variations in the basin-wide OT and its connection with ventilation of the open North Pacific remain unclear. In order to place our core results in a wider regional context, here, we compare our proxy records of sedimentary oxygenation (Uexcess concentration and Mo/Mn ratio) and export productivity (CaCO3) (Figures 6a, b, c) with

abundance of Pulleniatina obliquiloculata (an indicator of Kuroshio strength) and sea surface temperature (Shi et al., 2014), bulk sedimentary nitrogen isotope (an indicator of denitrification) (Kao et al., 2008), benthic foraminifera $\delta$13C (a proxy for water mass) in cores PN-3 and PC23A (Rella et al., 2012; Wahyudi and Minagawa, 1997), abundance of benthic foraminifera (an indicator of hypoxia) in core E017 (Li et al., 2005) and ODP167 site 1017 (Cannariato and Kennett, 1999) (Figures 6d-k)."

Lines 454-457: no, at those high temperatures you would only get a reduction in O2 of $\sim$ 3 for one degree warming, and 15 for a four degree warming (assuming no large salinity changes). Higher glacial salinity would cause less reduction in O2.

Reply#28: Corrected. The reference by Brewer and Peltzer (2016) has been included in the revised manuscript. Lines 469-473.

The sentence was changed to: "Based on thermal solubility effects, a hypothetical warming of 1°C would reduce oxygen concentrations by about 3.5 $\mu$mol/kg at water temperatures around 22°C (Brewer and Peltzer, 2016), ......"

Lines 457-458: sentence does not make sense. Reply#29:We have removed the sentence in the revised manuscript.

Lines 848-846ïijĹ484-486ïij§ïijĽ: does not make sense. How does subsurface water oxygen consumption lead to lower oxygen concentrations in deeper waters?

Reply#30: Thanks for your suggestion. We have removed the sentence in the revised manuscript. The replenishment of oxygen in deep water is controlled by both lateral advection and vertical supply. The oxygen consumption in subsurface water would reduce the oxygen concentration, thus lead to a less penetration of oxygen into deeper waters.

Lines 491-494: again not taking into account other factors that influence CaCO3 accumulation and preservation in sediments. For discussion on Kuroshio Current: see Lim et al. (2017).

Reply#31: Please see Reply #16 for CaCO3. Total Hg concentrations (Lim et al., 2017) has been invoked to explain variations in the intensity of Kuroshio. Interestingly, they concluded that the intensity of Kuroshio strengthened rapidly since 9.3 ka, whereas weakening and/or changing path of Kuroshio occurred during the last glaciation (20 ka - 9.3 ka). This conclusion further confirms weakening effect of Kuroshio on ventilation during the glacial period but an increase since 8.5 ka, consist with our inference.

We have added Lines 534-537. "More recently, lower hydrothermal total Hg concentration during 20 ka - 9.6 ka, associated with reduced intensity and/or variation in flow path of KC, relative to that of Holocene recorded in core KX12‐3 (1423 water depth) (Lim et al., 2017), further validates our inference."

Line 544-550: coined? Matsumoto et al. (2002) discuss one radiocarbon age from the Santa Barbara basin in relation to oxygen content, but at no point do they propose that GNPIW was stronger oxygenated. Cartapanis et al. (2011) and Ohkushi et al. (2013) discuss that the NE OMZ Pacific strengthened and weakened at millennial time scales, not glacial interglacial timescales. Also around the equatorial Pacific it is suggested that there was no difference in intermediate water oxygenation between the last glacial and Holocene (Hoogakker et al., 2018). Further down in the South Pacific Lu et al. (2016) suggest that upper waters were depleted in oxygen during the last glacial period.

Reply#32: In Matsumoto et al. (2002), published in Quaternary Science Reviews (2002, 21, 1693-1704), they used a compilation of benthic foraminiferal $\delta$13C to reveal the deep hydrography of the North Pacific. On page 1700 of this paper, they stated that "Although a water mass that reaches 2000 m should not be called intermediate water in the sense of modern physical oceanography, here we will refer to it as glacial NPIW (GNPIW) for the lack of a better name." According to our understanding, GNPIW refers to the nature of NPIW during cold intervals in the geological past, which is used to describe the state of NPIW at a variety of timescales, such as glacials, stadials, and Heinrich cold events. It should be noted that, since then, this term have been widely used in the literature related to NPIW (Cartapanis et al., 2011; Max et al., 2017; Worne

et al., 2019).

Cartapanis et al. (2011) stated in the abstract that "...... intermediate water oxygenation improved during H events, but slightly deteriorated during late Marine Isotope Stage (MIS) 3 and MIS 2." and they attributed to reorganization in regional intermediate oceanic circulation.

Ohkushi et al. (2013) revealed millennial-scale changes of OMZ of the NE Pacific with an increase at warming interval and a decrease at cold interval during the last deglaciation. They also found that "the middle to late Holocene (6–0 ka) was less dysoxic than the early Holocene."

Around the Eastern Equatorial Pacific, Hoogakker et al. (2018) suggested a persistent oxygen depletion of shallow depths during the last glacial period and the Holocene. Lu et al. (2016) concluded that the oxygen depletion in upper waters is closely related to poor ventilation and storage of respired carbon. Our data show increased ocean oxygenation at intermediate depth during the last glacial period and we attribute to enhanced NPIW formation at that time. Such spatial discrepancy in ocean oxygenation is ascribed to regional ocean circulation and export production.

Lines 550-556: generalised comment, what about brine (aka Kim et al. 2011).

Reply#33: Thanks. In the revised Manuscript, we shorten these sentences. Lines 568-570 in the revised MS.

The sentence now was changed to: "The intensified formation of GNPIW due to the displacement of source region to the Bering Sea was proposed by Ohkushi et al. (2003) and then is confirmed by Horikawa et al. (2010)."

Lines 556-559: what is intensified GNPIW?

Reply#34:"intensified GNPIW" means improved ventilation and deeper penetration of NPIW. Here, we also use enhanced NPIW formation to replace intensified GNPIW. Lines 575-576 in the revised MS.

The sentence was amended as follows. "......validate such inference, suggesting pronounced effects of intensified NPIW formation in the OT."

Discussion lines 560-571: needs tightening, it is unclear where this goes and how it relates to this study?

Reply#35: We have rewritten these sentences together with lines 572 - 581 (in the original manuscript). In the revised manuscript, the aim of this paragraph is to clarify the process of intensified GNPIW during HS1 and its substantial control on sedimentary oxygenation of the northern OT. Lines 577-592 in the revised MS.

We have changed the sentences as follows: "During HS1, a stronger formation of GN-PIW was supported by proxy studies and numerical simulations. For example, on the basis of paired benthic-planktic (B-P) 14C data, enhanced penetration of NPIW into a much deeper water depth during HS1 relative to the Holocene has been revealed in several studies (Max et al., 2014; Okazaki et al., 2010; Sagawa and Ikehara, 2008), which was also simulated by several models (Chikamoto et al., 2012; Gong et al., 2019; Okazaki et al., 2010). On the other hand, increased intermediate water temperature in the subtropical Pacific recorded in core GH08-2004 (1166 m water depth) (Kubota et al., 2015) and young deep water observed in the northern South China Sea during HS1 (Wan and Jian, 2014) along downstream region of NPIW are also related to intensified NPIW formation. Furthermore, the pathway of GNPIW from numerical model simulations (Zheng et al., 2016a) was similar to modern observations (You, 2003). Thus, all these evidence imply a persistent, cause and effect relation between GNPIW ventilation, the intermediate and deep water oxygen concentration in the OT and sediment redox state during HS1. In addition, our RSEs data also suggested a similarly enhanced ventilation in HS2 (Figures 6b and 6c) that is also attributed to intensified GNPIW."

Line 629: what is ocean ventilation seesaw? There is hardly any explanation for this in the main text, and Figure 7 shows strength of AMOC in the Atlantic and compares this

with the current study.

Reply#36: In the revised manuscript, the "ocean ventilation seesaw" has been replaced by "the mechanism behind such out-of-phase pattern between the ventilation in the subtropical North Pacific and the North Atlantic deep water formation" Lines 653-655 in the revised MS.

In addition, we also have added some discussion about the North Atlantic Climate in the 1st paragraph of section 6.3 and modified the text of the following sections to make it more logical.

Lines 625-636 "One of the characteristics climate features in the Northern Hemisphere, in particular the North Atlantic is millennial-scale oscillations during the glacial and deglacial periods. These abrupt climatic events have been widely thought to be related to varying strength of Atlantic Meridional Overturning Circulation (AMOC) (Lynch-Stieglitz, 2017). One of dynamic proxies of ocean circulation, 231Pa/230Th reveals that severe weakening of AMOC only existed during Heinrich stadials due to increased freshwater discharges into the North Atlantic (Böhm et al., 2015; McManus et al., 2004). On the other hand, several mechanisms, such as sudden termination of freshwater input (Liu et al., 2009), atmospheric CO2 concentration (Zhang et al., 2017), enhanced advection of salt (Barker et al., 2010) and changes in background climate (Knorr and Lohmann, 2007) were proposed to explain the reinvigoration of AMOC during the B/A."

Lines 637-638 "Our RSEs data in the Northern OT and epibenthic $\delta$13C in the Bering Sea (Figures 7a-c) both......"

Lines 653-655 "However, the mechanism behind such out-of-phase pattern between the ventilation in the subtropical North Pacific and the North Atlantic deep water formation remains unclear."

References Andres, M., Jan, S., Sanford, T. B., Mensah, V., Centurioni, L. R., and

Interactive
comment

Book, J. W.: Mean structure and variability of the Kuroshio from northeastern Taiwan to southwestern Japan, Oceanography, 26, 84–95, 2015. Böhm, E., Lippold, J., Gutjahr, M., Frank, M., Blaser, P., Antz, B., Fohlmeister, J., Frank, N., Andersen, M. B., and Deininger, M.: Strong and deep Atlantic meridional overturning circulation during the last glacial cycle, Nature, 517, 73-76, 2015. Barker, S., Knorr, G., Vautravers, M. J., Diz, P., and Skinner, L. C.: Extreme deepening of the Atlantic overturning circulation during deglaciation, Nature Geoscience, 3, 567-571, 2010. Beny, F., Toucanne, S., Skonieczny, C., Bayon, G., and Ziegler, M.: Geochemical provenance of sediments from the northern East China Sea document a gradual migration of the Asian Monsoon belt over the past 400,000 years, Quaternary Science Reviews, 190, 161-175, 2018. Bianchi, D., Dunne, J. P., Sarmiento, J. L., and Galbraith, E. D.: Data-based estimates of suboxia, denitrification, and N2O production in the ocean and their sensitivities to dissolved O2, Global Biogeochemical Cycles, 26, doi:10.1029/2011gb004209, 2012. Brewer, P. G. and Peltzer, E. T.: Ocean chemistry, ocean warming, and emerging hypoxia: Commentary, Journal of Geophysical Research: Oceans, 121, 3659-3667, 2016. Cannariato, K. G. and Kennett, J. P.: Climatically related millennial-scale fluctuations in strength of California margin oxygen-minimum zone during the past 60 k.y, Geology, 27, 975-978, 1999. Cartapanis, O., Tachikawa, K., and Bard, E.: Northeastern Pacific oxygen minimum zone variability over the past 70 kyr: Impact of biological production and oceanic ventilation, Paleoceanography, 26, PA4208, doi: 4210.1029/2011PA002126, 2011. Chang, A. S., Pedersen, T. F., and Hendy, I. L.: Effects of productivity, glaciation, and ventilation on late Quaternary sedimentary redox and trace element accumulation on the Vancouver Island margin, western Canada, Paleoceanography, 29, doi: 10.1002/2013PA002581, 2014. Chang, F., Li, T., Xiong, Z., and Xu, Z.: Evidence for sea level and monsoonally driven variations in terrigenous input to the northern East China Sea during the last 24.3‏ka, Paleoceanography, 30, 642-658, 2015. Chang, Y.-P., Chen, M.-T., Yokoyama, Y., Matsuzaki, H., Thompson, W. G., Kao, S.-J., and Kawahata, H.: Monsoon hydrography and productivity changes in the East China Sea during the past 100,000 years: Okinawa Trough evi-

dence (MD012404), Paleoceanography, 24, PA3208, doi: 3210.1029/2007PA001577, 2009. Cheng, H., Edwards, R. L., Sinha, A., Spötl, C., Yi, L., Chen, S., Kelly, M., Kathayat, G., Wang, X., Li, X., Kong, X., Wang, Y., Ning, Y., and Zhang, H.: The Asian monsoon over the past 640,000 years and ice age terminations, Nature, 534, 640-646, 2016. Chikamoto, M. O., Menviel, L., Abe-Ouchi, A., Ohgaito, R., Timmermann, A., Okazaki, Y., Harada, N., Oka, A., and Mouchet, A.: Variability in North Pacific intermediate and deep water ventilation during Heinrich events in two coupled climate models, Deep Sea Research Part II: Topical Studies in Oceanography, 61-64, 114-126, 2012. Clemens, S. C., Holbourn, A., Kubota, Y., Lee, K. E., Liu, Z., Chen, G., Nelson, A., and Fox-Kemper, B.: Precession-band variance missing from East Asian monsoon runoff, Nature Communications, 9, 3364, doi: 3310.1038/s41467-41018-05814-41460, 2018. Diekmann, B., Hofmann, J., Henrich, R. I., Futterer, D. K., Rohl, U., and Wei, K. Y.: Detrital sediment supply in the southern Okinawa Trough and its relation to sea-level and Kuroshio dynamics during the late Quaternary, Marine Geology, 255, 83-95, 2008. Dou, Y., Yang, S., Shi, X., Clift, P. D., Liu, S., Liu, J., Li, C., Bi, L., and Zhao, Y.: Provenance weathering and erosion records in southern Okinawa Trough sediments since 28ka: Geochemical and Sr–Nd–Pb isotopic evidences, Chemical Geology, 425, 93-109, 2016. Galbraith, E. D. and Jaccard, S. L.: Deglacial weakening of the oceanic soft tissue pump: global constraints from sedimentary nitrogen isotopes and oxygenation proxies, Quaternary Science Reviews, 109, 38-48, 2015. Gong, X., Lembke-Jene, L., Lohmann, G., Knorr, G., Tiedemann, R., Zou, J. J., and Shi, X. F.: Enhanced North Pacific deep-ocean stratification by stronger intermediate water formation during Heinrich Stadial 1, Nature Communications, 10, 656, doi:610.1038/s41467-41019-08606-41462, 2019. Hoogakker, B. A. A., Elderfield, H., Schmiedl, G., McCave, I. N., and Rickaby, R. E. M.: Glacial–interglacial changes in bottom-water oxygen content on the Portuguese margin, Nature Geoscience, 8, 40-43, 2015. Hoogakker, B. A. A., Lu, Z., Umling, N., Jones, L., Zhou, X., Rickaby, R. E. M., Thunell, R., Cartapanis, O., and Galbraith, E.: Glacial expansion of oxygen-depleted seawater in the eastern tropical Pacific, Nature, 562, 410-413, 2018. Horikawa, K., Asahara, Y., Yamamoto, K., and

[revised manuscript text omitted]
, 29, 1162-1178, 2014. Worne, S., Kender, S., Swann, G. E. A., Leng, M. J., and Ravelo, A. C.: Coupled climate and subarctic Pacific nutrient upwelling over the last 850,000 years, Earth and Planetary Science Letters, 522, 87-97, 2019. Wu, Y., Cheng, Z., and Shi, X.: Stratigraphic and carbonate sediment characteristics of Core CSH1 from the northern Okinawa Trough, Advances in Marine Science, 22, 163-169 (in Chinese with English Abstract), 2004. You, Y. Z.: The pathway and circulation of North Pacific Intermediate Water, Geophysical Research Letters, 30, doi:10.1029/2003gl018561, 2003. Zhang, X., Knorr, G., Lohmann, G., and Barker, S.: Abrupt North Atlantic circulation changes in response to gradual $CO_2$ forcing in a glacial climate state, Nature Geoscience, 10, 518-524, 2017. Zhao, D., Wan, S., Clift, P. D., Tada, R., Huang, J., Yin, X., Liao, R., Shen, X., Shi, X., and Li, A.: Provenance, sea-level and monsoon climate controls on silicate weathering of Yellow River sediment in the northern Okinawa Trough during late last glaciation, Palaeogeography, Palaeoclimatology, Palaeoecology, 490, 227-239, 2018. Zhao, D., Wan, S., Toucanne, S., Clift, P. D., Tada, R., Révillon, S., Kubota, Y., Zheng, X., Yu, Z., Huang, J., Jiang, H., Xu, Z., Shi, X., and Li, A.: Distinct control mechanism of fine-grained sediments from Yellow River and Kyushu supply in the northern Okinawa Trough since the last glacial, Geochemistry, Geophysics, Geosystems, 18, 2949-2969, 2017. Zheng, X., Kao, S., Chen, Z., Menviel, L., Chen, H., Du, Y., Wan, S., Yan, H., Liu, Z., Zheng, L., Wang, S., Li, D., and Zhang, X.: Deepwater circulation variation in the South China Sea since the Last Glacial Maximum, Geophysical Research Letters, 43, 8590-8599, 2016a. Zheng, X., Li, A., Kao, S., Gong, X., Frank, M., Kuhn, G., Cai, W., Yan, H., Wan, S., Zhang, H., Jiang, F., Hathorne, E., Chen, Z., and Hu, B.: Synchronicity of Kuroshio Current and climate system variability since the Last Glacial Maximum, Earth and Planetary Science Letters, 452, 247-257, 2016b. Zhu, A., Shi, X., Zou, J., Wu, Y., Zhang, H.,

and Bai, Y.: Sediment Provenance and Fluxes in the Northern Okinawa Trough During the last 88ka, Marine Geology & Quaternary Geology, 35, 1-8 (in Chinese with English Abstract), 2015. Zou, J., Shi, X., Liu, Y., Liu, J., Selvaraj, K., and Kao, S.-J.: Reconstruction of environmental changes using a multi-proxy approach in the Ulleung Basin (Sea of Japan) over the last 48 ka, Journal of Quaternary Science, 27, 891-900, 2012.

Captions Figure A1 Photomicrographs with Modular Stereo Microscope (Zeiss SteREO Discovery V12) show that both detrital and biogenic components of sediment coarse fraction (>63 $\mu$m) for 8.23 ka (120-124 cm), 9.26 ka (144-148cm), 10.98 ka (184-188 cm), 11.66 ka (200-204 cm), 12.92 ka (232-236 cm), 14.05 ka (264-268 cm), and 15.18 ka (296-300 cm) in core CSH1 at 200X magnification.

Figure 3. (a) Lithology and oxygen isotope ($\delta$18O) profile of planktic foraminifera species Globigerinoides ruber (G.ruber) in core CSH1. (b) Plot of ages versus depth for core CSH1. Three known ash layers are indicated by solid red rectangles. (c) Time series of linear sedimentation rate (LSR) from core CSH1. (d) Comparison of age model of core CSH1 with Chinese Stalagmite composite $\delta$18O curve of (Cheng et al., 2016). Tie points for CSH1 core chronology (Table 2) in Figures 3c and 3d are designated by colored crosses.

Figure 6. Proxy-related reconstructions of mid-depth sedimentary oxygenation at site CSH1 (this study) compared with oxygenation records from other locations of the North Pacific and published climatic and environmental records from the Okinawa Trough.

Please also note the supplement to this comment:
https://www.clim-past-discuss.net/cp-2019-70/cp-2019-70-AC1-supplement.zip

———————

Figure A1

[Figure]

**Fig. 1.** Figure A1

[Figure]

**Fig. 2.** Figure 3

**Fig. 3.** Figure 6

---

## Author Comment (AC2) · 26 Aug 2019

Zou et al., present a rather interesting study focusing on reconstructing the oxygenationhistory in the Okinawa Trough covering the last 50 kyrs. Specifically, the authors attempt to disentangle the typically confounding influence of export production (and by inference the oxygen consumption related to organic matter degradation) and bottom water ventilation on the sedimentary redox condition. Their geochemical records largely corroborate previous findings in that oxygenation patterns at intermediate depths in the North Pacific were primarily controlled by the production and ventilation of North

[Figure]

Pacific Intermediate Water. Specifically, conditions were generally better oxygenated during stadials, when NPIW was generally better ventilated and vertically expanded. Furthermore, their data support a general expansion of oxygen-depleted waters at intermediate depths during the B/A occupying large swaths of the North Pacific. The manuscript is well documented and quite detailed in places. The argumentation could be somewhat streamlined (see comments below) and would certainly benefit from editorial support. I would also recommend the argumentation to focus on the aspects outlined in the title and abstract.

Reply#1: We thank the reviewer for the recognition of significance of this study. Based on the suggestion of the reviewer, we revised manuscript thoroughly in revised mode with a focus on sedimentary oxygenation changes in the subtropical North Pacific and its linkage to the North Atlantic Climate (supplements 1 and 2). In the following, we address each specific point raised by the reviewer.

l. 35 – deep ocean carbon sequestration is certainly one the reasons potentially explaining lower glacial pCO2 concentrations, but certainly not the only one. Please rephrase to avoid unnecessary confusion

Reply#2: Thanks for your suggestion. Lines 32-34 in the revised MS.

We rephrased this sentence as the following. "Deep ocean carbon cycle, especially carbon sequestration and outgassing, is one of the competitive mechanisms to explain variations in atmospheric CO2 concentrations on orbital and millennial timescales."

l. 36 – I would suggest rephrasing as follows – However, the potential role of subtropical North Pacific subsurface waters in modulating.

Reply#3: Agreed and we have done so. Thanks. Lines 34-36.

The sentence is amended as follows. "However, the potential role of subtropical North Pacific subsurface waters in modulating atmospheric CO2 levels on millennial timescales is poorly constrained."

l. 48 and throughout – why is HS1 so much different from HS2 when considering their respective oxygenation history?

Reply#4: During HS1 and HS2, our RSEs suggest enhanced sedimentary oxygenation due to intensified GNPIW. As suggested by the Reviewer, a slight difference in the structure of ventilation mode can be observed (Figure 7). We think this slight discrepancy could be related to different climatic background during the formation of HS1 and HS2. Records from paleoclimate archives, such as ice cores and Chinese cave stalagmites, show differences in structure, duration, amplitude between HS1 and HS2. Such differences are thought to be related to the climate background state, such as, CO2 concentration, ice sheet volume, AMOC intensity, sea ice extent and source of freshwater, etc.(Flückiger et al., 2006; Hemming, 2004; Kaspi et al., 2004; Lynch-Stieglitz et al., 2014) The response of NPIW to HS1 and HS2 events could be different, and thus cause a slight difference in sedimentary oxygenation. On the other hand, the discrepancy in export productivity (CaCO3) during these two cold intervals at core CSH1 could also play a role in controlling sedimentary oxygenation.

l. 62 – agreed. But I would add that in addition of the flushing of a poorly ventilated deep water mass upon the resumption of NADW, many export production records show a drastic increase during the B/A (e.g. Kohfeld& Chase, 2011), which could account for enhance oxygen removal associated with organic matter respiration upstream of the core site location.

Reply#5: Agreed with your suggestion and revised. Line 52.

The sentence is changed to "......due to upwelling of aged, nutrient-rich deep water and enhanced export production......"

l. 70 – AT the sediment-water interface

Reply#6: We reworded the sentence. Lines 62-66.

The sentence is amended as follows. "The sluggish ocean ventilation and efficient biological pump in the ocean facilitate carbon sequestration in the ocean interior, linking to atmospheric $CO_2$ drawdown, which in turn play a crucial role in regulating sedimentary oxygen on millennial and orbital timescales (Hoogakker et al., 2015; Jaccard and Galbraith, 2012; Sigman and Boyle, 2000)."

l. 83 . . . in marine sediment cores

Reply#7: Corrected. Line 75.

The sentence is amended as follows. "......in marine sediment cores."

l. 86 . . . in the subarctic North Pacific.

Reply#8: Corrected. Line 78.

The sentence is amended as follows. "...... last glaciation and Holocene in the subarctic North Pacific."

l. 92 – I would suggest to briefly explain what cabbeling means

Reply#9:Done. Cabbeling is a mixing process to form a new water mass with increased density than that of parent water masses.

We have added in Lines 85-86. "......that cabbeling, a mixing process to form a new water mass with increased density than that of parent water masses, is the principle mechanism responsible......"

l. 96 and throughout (incl. Fig. 6) – this may well be a semantic issue, but benthic d13C cannot be considered as a ventilation proxy per se, as the isotopic value can be obfuscated by air-sea gas exchange in locations where subsurface water masses form.

Reply#10: We fully agreed with the comment by the Reviewer that temporal benthic $\delta$13C changes is influenced by a variety of factors, such as air-sea equilibration, ocean-circulation changes, and productivity changes, etc. Previous studies also revealed benthic $\delta$13C pattern at basin-wide scales may reflect ocean-circulation changes (Charles

and Fairbanks, 1992; Charles et al., 1996; Ninnemann and Charles, 2002). In this study we also noticed similar trends in benthic $\delta$13C during $\sim$22 ka and $\sim$14 ka between cores PN-3 (Okinawa Trough) and PC23A (Bering Sea) (Figure 4), despite their great distance. In the revised manuscript, we add additional information for benthic $\delta$13C. The sentence was amended Lines 90-93. "Benthic foraminiferal $\delta$13C, a quasi-conservative tracer for water mass, from the North Pacific suggested an enhanced ventilation (enriched $\delta$13C) at water depths of < 2000 m during the last glacial period (Keigwin, 1998; Matsumoto et al., 2002)."

l. 151 – processes that are related to the supply of oxygen by ocean circulation and.

Reply#11: Revised. Lines 151-154. The sentence is changed to: "Sedimentary redox condition is governed by the rate of oxygen supply from the overlying bottom water and the rate of oxygen removal from pore water (Jaccard et al., 2016), processes that are related to the supply of oxygen by ocean circulation and organic matter respiration, respectively."

l. 163 – technically one should specify that under reducing conditions it is the authigenic fraction of Mn (as opposed to its detrital background) that remains low.

Reply#12: Revised. Lines 163-164.

The sentence was amended as follows: "Under reducing conditions, the authigenic fraction of Mn (as opposed to its detrital background) ......"

l. 168 – please add adequate reference

Reply#13:The reference (Morford and Emerson 1999) has been included in the revised manuscript. Line 171.

l. 195 – volcanism

Reply#14: Revised. Line 199.

"volcanic " has been changed to "volcanism".

§3 – I would suggest to substantially shorten this paragraph as the general oceanographic setting is already outlined in the introduction. I would recommend focusing on the aspects directly relevant to the argumentation (nutrient, dissolved O2).

Reply#15: Thanks for your suggestion. In the revised manuscript, we removed the 1st paragraph of section 3 and then we reworded the sentences in previous 3rd paragraph of section 3. Lines 210-228.

The paragraph is amended as follows. "Figures 2a and 2b show that the lower sea surface salinity (SSS) zone in summer relative to the one in winter in the ECS migrates toward the east of OT, indicating enhanced impact of the Changjiang discharge associated with summer EAM. An estimated ∼80% of the mean annual discharge of Changjiang is supplied to the ECS (Ichikawa and Beardsley, 2002) and in situ observational data show a pronounced negative correlation between the Changjiang discharge and SSS in July (Delcroix and Murtugudde, 2002). Consistently, previous studies from the OT reported such close relationship between summer EAM and SSS back to the late Pleistocene (Chang et al., 2009; Clemens et al., 2018; Kubota et al., 2010; Sun et al., 2005)."

l. 287 – why is the sedimentation rate so high during HS2 when both export production and detrital input (based on Al) are low? I would suggest verifying the age pointers for that interval.

Reply#16: Thanks for your suggestion and we have verified the age control points (Figure 3 and Table 2). In the original manuscript, higher sedimentation rate mainly occurred during HS2 and HS3 (>40-60 cm/ka) and is mainly caused by uncertainties of age control points at 23.476 ka (DO2) and 29.995 ka (H3). In the revised manuscript, these two age control points have been eliminated. Even with this more conservative tuning approach, the conclusions remain the same as before and robust.

l.291 – 85 samples covering 50 kyrs cannot provide an average time resolution of 200 yrs.

Reply#17: We thank the reviewer to point out this mistake. Now we have corrected it to ∼ 600 years. Line 289.

The sentence was changed to: "......with time resolution of about 600 years (every 4 cm interval)"

l. 345-346 – Maybe. But it may also suggest that the sedimentary CaCO3 content could be directly controlled by dilution. I would interpret the export productivity records with caution.

Reply#18: We thank the reviewer for the very helpful comments and suggestions. In the revised manuscript, we have discussed the effects of some factors, such as dilution and dissolution, on CaCO3. On the other hand, we have reworded this section to cautiously interpret CaCO3 as a reliable proxy for export production at core CSH1.

We have added the following information to Lines 342-361. "Several lines of evidence support CaCO3 as a reliable productivity proxy, particularly during the last deglaciation. The strong negative correlation coefficient ($r = -0.85$, p<0.01) between Al and CaCO3 in sediments throughout core CSH1 confirms the biogenic origin of CaCO3 against terrigenous Al (Figure 4f). Generally, terrigenous dilution decreases the concentrations of CaCO3. Inconsistent relationship between percentage CaCO3 and sedimentation rate indicates a minor effect of dilution on CaCO3. Furthermore, the increasing trend in CaCO3 associated with high sedimentation rate during the last deglacial interval indicates a substantial increase in export productivity (Figures 4a and 4d). The high coherence between percent CaCO3 and alkenone-derived sea surface water (SST) (Shi et al., 2014) indicates a direct control on CaCO3 by SST. Moreover, a detailed comparison between CaCO3 concentrations and the previously published foraminiferal fragmentation ratio (Wu et al., 2004) shows, apart from a small portion within the LGM, no clear co-variation between them. These evidence suggest that CaCO3 changes are driven primarily by variations in carbonate primary production, and not overprinted by secondary processes, such as carbonate dissolution through changes in the lysocline

depth and dilution by terrigenous materials. Likewise, similar deglacial trend in CaCO3 is also observed in core MD01-2404 (Chang et al., 2009), indicating a ubiquitous, not local picture in the OT. All these lines of evidence thus support CaCO3 of core CSH1 as a reliable productivity proxy to a first order approximation."

l.371-372 – interesting idea!

Reply#19:Thanks.

l. 427 (438?) – supply seems more adequate than provision.

Reply#20: Agreed and revised. Line 446.

"provision" was replaced by"supply".

l. 487-488 – please also consider citing Galbraith et al., 2007. Reply#21:The paper has been included in the revised manuscript. Line 502.

l. 489-490 – please keep in mind that O2 can be consumed upstream of the core site location as the removal of O2 in relation to organic matter degradation is integrated over the flow path of a give subsurface water mass.

Reply#22: Thanks for your suggestion. We fully agree with the comment and think that the signal of accumulation of organic matter degradation along the flow can be recorded by epibenthic $\delta$13C.

Fig. 6C – shouldn't the grey triangle right of the vertical axis be flipped upside down (i.e. high Mo/Mn coincident with low oxygenation)?

Reply#23: Thanks. High Mo/Mn ratio indicates low oxygen conditions.

The sign in Fig. 6c has been corrected (see new Figure 6).

References Chang, Y.-P., Chen, M.-T., Yokoyama, Y., Matsuzaki, H., Thompson, W. G., Kao, S.-J., and Kawahata, H.: Monsoon hydrography and productivity changes in the East China Sea during the past 100,000 years: Okinawa Trough evidence (MD012404),

Paleoceanography, 24, PA3208, doi: 3210.1029/2007PA001577, 2009. Charles, C. D. and Fairbanks, R. G.: Evidence from Southern Ocean sediments for the effect of North Atlantic deep-water flux on climate, Nature, 355, 416-419, 1992. Charles, C. D., Lynch-Stieglitz, J., Ninnemann, U. S., and Fairbanks, R. G.: Climate connections between the hemisphere revealed by deep sea sediment core/ice core correlations, Earth and Planetary Science Letters, 142, 19-27, 1996. Cheng, H., Edwards, R. L., Sinha, A., Spötl, C., Yi, L., Chen, S., Kelly, M., Kathayat, G., Wang, X., Li, X., Kong, X., Wang, Y., Ning, Y., and Zhang, H.: The Asian monsoon over the past 640,000 years and ice age terminations, Nature, 534, 640-646, 2016. Clemens, S. C., Holbourn, A., Kubota, Y., Lee, K. E., Liu, Z., Chen, G., Nelson, A., and Fox-Kemper, B.: Precession-band variance missing from East Asian monsoon runoff, Nature Communications, 9, 3364, doi: 3310.1038/s41467-41018-05814-41460, 2018. Delcroix, T. and Murtugudde, R.: Sea surface salinity changes in the East China Sea during 1997–2001: Influence of the Yangtze River, Journal of Geophysical Research: Oceans, 107, 8008, doi:8010.1029/2001JC000893, 2002. Flückiger, J., Knutti, R., and White, J. W. C.: Oceanic processes as potential trigger and amplifying mechanisms for Heinrich events, Paleoceanography, 21, 2006. Hemming, S. R.: Heinrich events: Massive late Pleistocene detritus layers of the North Atlantic and their global climate imprint, Reviews of Geophysics, 42, 2004. Hoogakker, B. A. A., Elderfield, H., Schmiedl, G., McCave, I. N., and Rickaby, R. E. M.: Glacial–interglacial changes in bottom-water oxygen content on the Portuguese margin, Nature Geoscience, 8, 40-43, 2015. Ichikawa, H. and Beardsley, R. C.: The Current System in the Yellow and East China Seas, Journal of Oceanography, 58, 77-92, 2002. Jaccard, S. L. and Galbraith, E. D.: Large climate-driven changes of oceanic oxygen concentrations during the last deglaciation, Nature Geoscience, 5, 151-156, 2012. Jaccard, S. L., Galbraith, E. D., Martínez-García, A., and Anderson, R. F.: Covariation of deep Southern Ocean oxygenation and atmospheric CO2 through the last ice age, Nature, 530, 207-210, 2016. Kaspi, Y., Sayag, R., and Tziperman, E.: A "triple sea-ice state" mechanism for the abrupt warming and synchronous ice sheet collapses during Heinrich events, Paleoceanography, 19, 2004.

Keigwin, L. D.: Glacial-age hydrography of the far northwest Pacific Ocean, Paleoceanography, 13, 323-339, 1998. Kubota, Y., Kimoto, K., Tada, R., Oda, H., Yokoyama, Y., and Matsuzaki, H.: Variations of East Asian summer monsoon since the last deglaciation based on Mg/Ca and oxygen isotope of planktic foraminifera in the northern East China Sea, Paleoceanography, 25, PA4205, doi:4210.1029/2009pa001891, 2010. Lynch-Stieglitz, J., Schmidt, M. W., Gene Henry, L., Curry, W. B., Skinner, L. C., Mulitza, S., Zhang, R., and Chang, P.: Muted change in Atlantic overturning circulation over some glacial-aged Heinrich events, Nature Geoscience, 7, 144, 2014. Matsumoto, K., Oba, T., Lynch-Stieglitz, J., and Yamamoto, H.: Interior hydrography and circulation of the glacial Pacific Ocean, Quaternary Science Reviews, 21, 1693-1704, 2002. Ninnemann, U. S. and Charles, C. D.: Changes in the mode of Southern Ocean circulation over the last glacial cycle revealed by foraminiferal stable isotopic variability, Earth and Planetary Science Letters, 201, 383-396, 2002. Shi, X., Wu, Y., Zou, J., Liu, Y., Ge, S., Zhao, M., Liu, J., Zhu, A., Meng, X., Yao, Z., and Han, Y.: Multiproxy reconstruction for Kuroshio responses to northern hemispheric oceanic climate and the Asian Monsoon since Marine Isotope Stage 5.1 (∼88 ka), Climate of the Past, 10, 1735-1750, 2014. Sigman, D. M. and Boyle, E. A.: Glacial/interglacial variations in atmospheric carbon dioxide, Nature, 407, 859-869, 2000. Sun, Y. B., Oppo, D. W., Xiang, R., Liu, W. G., and Gao, S.: Last deglaciation in the Okinawa Trough: Subtropical northwest Pacific link to Northern Hemisphere and tropical climate, Paleoceanography, 20, PA4005, doi:4010.1029/2004pa001061, 2005. Wu, Y., Cheng, Z., and Shi, X.: Stratigraphic and carbonate sediment characteristics of Core CSH1 from the northern Okinawa Trough, Advances in Marine Science, 22, 163-169 (in Chinese with English Abstract), 2004.

Captions Figure 3. (a) Lithology and oxygen isotope ($\delta$18O) profile of planktic foraminifera species Globigerinoides ruber (G.ruber) in core CSH1. (b) Plot of ages versus depth for core CSH1. Three known ash layers are indicated by solid red rectangles. (c) Time series of linear sedimentation rate (LSR) from core CSH1. (d) Comparison of age model of core CSH1 with Chinese Stalagmite composite $\delta$18O curve of (Cheng et al., 2016). Tie points for CSH1 core chronology (Table 2) in Figures 3c and

3d are designated by colored crosses.

Figure 6. Proxy-related reconstructions of mid-depth sedimentary oxygenation at site CSH1 (this study) compared with oxygenation records from other locations of the North Pacific and published climatic and environmental records from the Okinawa Trough.

Please also note the supplement to this comment:
https://www.clim-past-discuss.net/cp-2019-70/cp-2019-70-AC2-supplement.zip

[Figure]

[Figure]

**Fig. 1.** Figure 3

**Fig. 2.** Figure 6

---

## Author Response (AR1)

**Point-by-point response to referee comments**

**Response to reviewer**

Key:

- Reviewers' comments

- Author's response

- Modified text in the manuscript

**Anonymous Referee #1**

Review Zou et al 'Millennial-scale variations of sedimentary oxygenation in the westernsubtropical North Pacific and its link to North Atlantic climate.

Zou et al. study sedimentary redox conditions in the Okinawa Trough, and use this as a proxy to infer bottom water oxygen concentrations (I think, it is not always very clear from the manuscript). The title promises more than the paper delivers; the link with North Atlantic climate is only mentioned briefly and explanation is sometimes unclear.

The authors present interesting data, but the paper itself needs work. There is so much information (several times incorrectly referenced), and some information seems irrelevant. There is also a lot of internal discussion within the paper without reaching firm conclusions. While the authors are critical about their own proxy, they are less so about others and this needs to be improved.

**Reply#1:** We thank Reviewer#1 for taking the time to review our manuscript and for the constructive comments, which contribute improving our manuscript. Our work adds an important element to understand the process in the subtropical North Pacific during the last deglaciation. Our data suggest a substantial impact of NPIW on sedimentary oxygenation in the western subtropical North Pacific and also an expansion of oxygen minimum zone in the North Pacific during the B/A.

According to the comments of Reviewers #1 and #2, we have carried out a thorough revision of our entire manuscript, including Figures 3-7, and use the "track changes" to display our revisions the text. In the revised manuscript, we re-phrased the abstract, amended the age model (see new Figure 3), changed sentences and paragraphs as outlined in the revised text, added new evidence of benthic foraminifera abundance of core E017 (see new Figure 6) and added text to the discussion on the connection of our time series to North Atlantic climate. We hope that we have addressed Reviewer #1' concerns appropriately.

In the following we provide point-by-point responses to all the Reviewer's comments in blue as well as excerpts from the manuscript in green.

Comments: Authors should check that their references are appropriate. Several references are not put in the right context.

**Reply#2:** Thanks. Checked and Revised.

**The following references were included** in the revised MS**:**

in Line 66 for reference Hoogakker et al., 2015

in Line 75 for reference Addison et al., 2012

in Lines 100 for reference Max et al 2017; Rippert et al. 2017

in Line 107 for reference Sibuet et al., 1987

in Line 170 for reference Morford and Emerson, 1999

in Line 333 for references Cartapanis et al., 2011; Lembke-Jene et al., 2017

in Line 348 for reference Shi et al., 2014

in Line 356 for reference Chang et al., 2009

in Line 368 for reference Zhu et al., 2015

in Line 372 for reference Lim et al., 2017

in line 396 for reference Bianchi et al. 2012

in lines 444-445 for references Galbraith and Jaccard, 2015; Moffitt et al., 2015; Praetorius et al., 2015

in line 465 for reference Brewer and Peltzer, 2016

in line 496 for reference Galbraith et al., 2007

in line 530 for reference Lim et al., 2017

in line 535 for reference Andres et al., 2015

in line 551 for reference Kubota et al., 2015

in line 578 for reference Kubota et al., 2015

in line 623 for reference Lynch-Stieglitz, 2017

in lines 625-630 for references Böhm et al., 2015; McManus et al., 2004; Liu et al., 2009; Zhang et al., 2017; Barker et al., 2010; Knorr and Lohmann, 2007; in lines 697-698 for references Galbraith and Jaccard, 2015; Jaccard and Galbraith, 2012; Moffitt et al., 2015

in lines 709-710 for references Addison et al., 2012; Cartapanis et al., 2011; Crusius et al., 2004; Galbraith et al., 2007; Lembke-Jene et al., 2017; Shibahara et al., 2007

**The following original references were removed** in the revised MS:

in line 65 Lu et al., 2016; in line 400 Savrda and Bottjer, 1991

in line 471 Benson and Krause, 1984

in line 541 Matsumotoet al. (2002)

in line 557 Kubota et al., 2010

in line 565 Wang and Wang, 2008

in line 584 Kubota et al., 2010

There are several other studies that deal with the North Pacific and NPIW, which are not referenced here; this includes work by Rippert et al. (2017), Max et al. (2017).

**Reply#3:** Thanks. These two references have been included in the revised manuscript and we briefly introduced their main findings.

In particular, both these papers highlight the substantial effects of NPIW on subsurface water composition of Eastern Tropical North Pacific and potent roles in regulating global climate. Likewise, the effect of NPIW's ventilation on the western subtropical North Pacific is observed in our study. Our study further validates the role of NPIW in its downstream oceanic environment.

We have added Lines 97-100

"In contrast, substantial effects of intensified NPIW formation during Marine Isotope Stage (MIS) 2 and 6 on the ventilation and nutrient characteristics of lower latitude mid-depth Eastern Equatorial Pacific have been suggested by recent studies (Max et al., 2017; Rippert et al., 2017)."

Abstract: Lines 44-50: these sentences go around the bushes. Really what you want to say is that sedimentary oxygenation conditions at mid-depth in the subtropical western North Pacific were more or less similar over the last 50,000 years, apart from the Bolling-Allerod and Pre-boreal. However, it may not be possible to compare with Holocene data, as this may be compromised by ash. This is not made very clear in the manuscript.

**Reply#4:** Thanks for your suggestion. We have rephrased these sentences in the revised manuscript. The sentence was amended as follows. The text was modified as follows (Lines 42-45):

"Our results suggest that enhanced mid-depth western subtropical North Pacific (WSTNP) sedimentary oxygenation occurred during cold intervals and after 8.5 ka, while oxygenation decreased during the Bölling-Alleröd (B/A) and Preboreal. "

As suggested by the Reviewer, our data suggest well-oxygenated water during the cold intervals apart from the B/A and Preboreal. For the Holocene, increased sedimentary oxygenation is attributed to an intensified Kuroshio Current, although discrete volcanic materials, indicating by positive Eu anomaly (Zhu et al., 2015), would dilute the Holocene data. Modern observation suggest the Kuroshio can reach to the seafloor at 1200 m in isobath in the East China Sea (Andres et al., 2015). In the geological past, both various proxy data and modeling simulation suggest intensified Kuroshio re-entered the OT at early Holocene 9-9.6 ka (Chang et al., 2015; Diekmann et al., 2008; Dou et al., 2016; Lim et al., 2017; Zheng et al., 2016). In our previous study based on the same core CSH1 (Zhu et al., 2015), we have suggested that the occurrence of discrete volcanic materials during the Holocene is closely related to enhanced Kuroshio intensity. More recently, increased total Hg concentration in the sediments from the middle Okinawa since 9.3 ka (Lim et al., 2017) was also suggested, which was explained to be hydrothermal Hg source (due to much higher concentrations than potential terrigenous end-members) and to be brought to the site location by intensified Kuroshio Current. Although the focus of this manuscript is not the ventilation changes during the Holocene, the Kuroshio Current does play a crucial role in controlling the ventilation of the OT during the Holocene.

We have now added more information into the section 5.2 Redox-sensitive Elements. Lines 366-372:

"Pronounced variations in U concentration after 8.5 ka are related to the occurrence of discrete volcanic materials. A significant positive Eu anomaly (Zhu et al., 2015) confirms the occurrence of discrete volcanic materials and its dilution effects on terrigenous components since 7 ka. Occurrence of discrete volcanic material is likely related to intensified Kuroshio Current during the mid-late Holocene, as supported by higher hydrothermal Hg concentrations in sediments from the middle OT (Lim et al., 2017)."

Lines 59-61: how does it seem to be driven? Is this not something you are proposing? Then it is not seem.

**Reply#5:** Thanks. Revised. "seem to be driven" is replaced by "was likely driven". Lines 48-51 in the revised MS. The sentence now reads:

"The enhanced formation of NPIW during Heinrich Stadial 1 (HS1) was likely driven by the perturbation of sea ice formation and sea surface salinity oscillations in high-latitude North Pacific."

The authors mix up NADW and the Atlantic Meridional Overturning Circulation. For a good description of AMOC see recent paper by Frajke-Williams et al. (2019).

**Reply#6:** Thanks. We checked the paper by Frajke-Williams et al. (2019). In the revised manuscript, we replaced AMOC by North Atlantic Deep Water and reworded these sentences. Lines 51-57 in the revised MS. The sentences now read:

"The diminished sedimentary oxygenation during the B/A due to decreased NPIW formation and enhanced export production, indicates an expansion of oxygen minimum zone in the North Pacific and enhanced $CO_2$ sequestration at mid-depth waters, along with termination of atmospheric $CO_2$ concentration increase. We attribute the millennial-scale changes to intensified NPIW and enhanced abyss flushing during deglacial cold and warm intervals, respectively, closely related to variations in North Atlantic Deep Water formation."

Introduction: Lines 70-73: not sure how to interpret this. Where is the respired carbon stored? At the sediment-seawater interface, in sedimentary pore-waters, or in seawater? The study of Lu et al. (2016) deals with I/Ca in planktic foraminifer in the Pacific Sector of the Southern Ocean, to reconstruct upper ocean oxygenation, the part of respired carbon in their paper refers to a different study (Hoogakker et al., 2015).

**Reply#7:** We have reworded the first sentence for a clearer statement. Additionally, the reference (Lu et al., 2016) was replaced by reference (Hoogakker et al., 2015).

The sentence was amended as follows, Lines 61-66:

"A more sluggish deep ocean ventilation combined with a more efficient biological pump widely thought to facilitate enhanced carbon sequestration in the ocean interior, leading to atmospheric $CO_2$ drawdown during glacial cold periods (Sigman and Boyle, 2000). These changes are tightly coupled to bottom water oxygenation and sedimentary redox changes on both millennial and orbital timescales (Hoogakker et al., 2015; Jaccard and Galbraith, 2012; Sigman and Boyle, 2000)."

Lines 76-83: the study of Cartapanis is from the northeastern Pacific, but not high latitude or subarctic.

**Reply#8:** The study by Cartapanis et al. (2011) presents high-resolution redox-sensitive trace metals in a sediment core from the Eastern Tropical North Pacific and proposed improved intermediate water oxygenation during Heinrich events. The enhanced ventilation during Heinrich intervals is consistent with our inference. Therefore, we would like to continue to keep this reference in the revised manuscript and rephrased the expression of the context in Lines 69-72. Now the sentence reads:

"Previous studies from North Pacific margins as well as open subarctic Pacific have identified drastic variations in export productivity and ocean oxygen levels at millennial and orbital timescales using diverse proxies such as trace elements (Cartapanis et al., 2011; Chang et al., 2014; Jaccard et al., 2009; Zou et al., 2012), ......"

Line 92: explain what cabbeling is, not everyone will have heard of the term.

**Reply#9:** Done. Cabbeling is a mixing process to form a new water mass with increased density than that of parent water masses. Lines 85-86 in the revised MS. The sentence was changed to:

"...... that cabbeling, a mixing process to form a new water mass with increased density than that of parent water masses, is the principle mechanism responsible for ......"

Lines 95- 97: do the data really show this? The one core at 1km is about 0.04 per mil lighter (and within error), but crucially there is no Holocene equivalent for the 0.7 km core.

**Reply#10:** These two studies show enhanced formation of North Pacific Intermediate Water during the last glaciation. On the basis of >30 sediment cores on the northern Emperor Seamounts and in the Okhotsk Sea with a water depth of 1000 to 4000 m, Keigwin (1998) found that there was a better ventilated and relatively fresher water mass above 2000 m in the far northwestern Pacific compared to deep waters. Matsumoto et al. (2002) compiled and compared available nutrient proxies ($\delta^{13}$C) in sediment cores with water depth of 740 m to 3320 m in the North Pacific and found a presence of distinctive water masses below and above 2000 m water depth in the glacial Pacific with higher benthic $\delta^{13}$C in the upper 2000 m water.

Line2 149-152: do you mean 'is governed by' instead of 'is the balance between'.

**Reply#11:** Revised. Lines 150-153 in the revised MS. Now the sentence reads:

"The sedimentary redox conditions are governed by the rate of oxygen supply from the overlying bottom water and the rate of oxygen removal from pore water (Jaccard et al., 2016), processes that are related to the supply of oxygen by ocean circulation and organic matter respiration, respectively."

Figure 1. O2 map, and locations of cores. Are all the cores discussed in the paper? Is it worrying that the main core CSH1 is from just south of Japan and perhaps should not be considered an open ocean core? What do the letters A to E stand for?

**Reply#12:** All cores shown in Figure 1b have been discussed in this manuscript. For example, benthic $\delta^{13}C$ in core PN-3 was used to indicate ventilation change in the OT. Concentrations of $CaCO_3$ and reactive phosphorus recorded in core MD01-2404 were used to correlate with our productivity proxy. Benthic foraminiferal assemblages in cores E017 and 225 retrieved from the middle and southern OT were used to indicate the ventilation of deep water mass in the OT. Deep-water temperatures in cores GH08-2004 and GH02-1030, benthic $\delta^{13}C$ in core PC23A and benthic foraminiferal assemblages in ODP Site 1017 have implicated the ventilation of North Pacific Intermediate Water.

Core CSH1 is situated in the northern Okinawa Trough at a water depth of 703 m. In this area, both surface and deep water can be continuously replenished by water masses from open ocean.

Letters A to E stand for sediment cores previously reported in and near the Okinawa Trough and are shown in Table 1.

Setting: Do details of discharge and SSS add anything to this study?

**Reply#13:** Sea surface salinity (SSS) in the East China Sea is closely related to the intensity of the summer East Asian Monsoon (EAM). On the basis of this relationship, a recent study from the northern Okinawa Trough (U1429) has extended this relationship back to 400 ka (Clemens et al., 2018). In order to discern millennial-scale variability, we correlate planktic $\delta^{18}O$ of core CSH1 with Chinese stalagmite $\delta^{18}O$, an indicator of summer EAM to establish the age model for core CSH1 in our study. We would like to add the details of discharge and SSS in the manuscript to let the readers know the details.

Material and methods: What causes the high accumulation rates in this core? As the accumulation rates vary significantly, how do these influence the patterns in redox elements etc?

**Reply#14:** In the northern Okinawa Trough, previous sediment provenance studies suggested the terrigenous sediments are mainly sourced from the Yellow and Changjiang Rivers, from China and short mountainous rivers (Japanese or even

Taiwanese rivers) from the surrounding islands, as well as eolian dust, volcanic and hydrothermal materials from Yellow River, Changjiang and part of Korea and Japan Rivers (Beny et al., 2018; Li et al., 2015; Zhao et al., 2018; Zhao et al., 2017). Variation in sedimentation rate has been attributed to changes in eustatic sea level, EAM intensity, path and/or intensity of Kuroshio Current. Generally, sea level is thought to be the first-order factor for controlling linear sedimentation rate changes (Beny et al., 2018; Li et al., 2015; Zhao et al., 2017).

Our data show that there is no coherent relationship between linear sedimentation rate and concentrations of redox sensitive elements. For example, high sedimentation rate between 24.2 ka and 32.4 ka (around 40 cm/ka) corresponds to decreasing concentrations of redox sensitive elements. On the other hand, lower sedimentation between 16 ka and 19.8 ka also corresponds to lower concentrations of redox sensitive elements. Therefore, linear sedimentation rate is not deemed to be a crucial factor in controlling concentrations of redox sensitive elements in core CSH1.

Lines 339-340: preservation of TOC and CaCO3 are influenced by many factors and not a widely used paleo-export proxy.

**Reply#15:** We fully agree that preservation of TOC and $CaCO_3$ are influenced by many factors, including supply, dissolution, organic matter degradation, terrigenous dilution, etc. Some factors can be ruled out and at times these two proxies have been used to reconstruct export productivity. In this study, C/N molar shows substantial contribution of terrigenous organic matter to total organic carbon, therefore it is not a suitable proxy for productivity reconstruction.

In the revised manuscript, we showed multiple lines of evidence to support the utility of $CaCO_3$ as a reliable productivity proxy, including (1) a strong negative correlation with terrigenous Al of core CSH1; (2) weak dilution effects of terrigenous material on $CaCO_3$; (3) similar pattern to sea surface temperature of core CSH1 (Shi et al., 2014) (the data have been included in Figure 6); (4) similar deglacial trends in $CaCO_3$ and reactive phosphorus reported in core MD012404 retrieved from the middle OT (Chang et al., 2009; Li et al., 2017). All these lines of evidence support $CaCO_3$ as a proxy for productivity in the study area. We have added these information in lines 339-358 in the revised MS:

"Several lines of evidence support $CaCO_3$ as a reliable productivity proxy, particularly during the last deglaciation. The strong negative correlation coefficient (r = − 0.85, p<0.01) between Al and $CaCO_3$ in sediments throughout core CSH1 confirms the biogenic origin of $CaCO_3$ against terrigenous Al (Figure 4f). Generally, terrigenous dilution decreases the concentrations of $CaCO_3$. An inconsistent relationship between $CaCO_3$ contents and sedimentation rates indicates a minor effect of dilution on $CaCO_3$. Furthermore, the increasing trend in $CaCO_3$ associated with high sedimentation rate during the last deglacial interval indicates a substantial increase in export productivity (Figures 4a and d). The high coherence between $CaCO_3$ content and alkenone-derived sea surface water (SST) (Shi et al., 2014) indicates a direct control on $CaCO_3$ by SST. Moreover, a detailed comparison between $CaCO_3$ concentrations and the previously published foraminiferal fragmentation ratio (Wu et al., 2004) shows, apart from a small portion within the LGM, no clear co-variation between them. These evidence suggest that $CaCO_3$ changes are driven primarily by variations in carbonate primary production, and not overprinted by secondary processes, such as carbonate dissolution through changes in the lysocline depth and dilution by terrigenous materials. Likewise, a similar deglacial trend in $CaCO_3$ is also observed in core MD01-2404 (Chang et al., 2009), indicating a ubiquitous, not local picture in the OT. All these lines of evidence thus support $CaCO_3$ of core CSH1 as a reliable productivity proxy to a first order approximation."

Line 354-357: have you checked that it is an extant biological component that makes up the high CaCO3 going from B/A to ~ 8000 years? Can you explain the differences in the LSR figures between Figures 3 and 4? For examples, in Figure 3 highest rates occur centred around 22 kyrs as part of a large interval of high LSR (from 30 to 20), whereas in Figure 4 this occurs earlier (33 to 24 kyrs).

**Reply#16:** Seven samples at 8.23ka (120-124cm), 9.26ka (144-148cm), 10.98ka (184-188cm), 11.66ka (200-204cm), 12.92ka (232-236cm), 14.05ka (264-268cm), and 15.18ka (296-300cm) in core CSH1 were analyzed by Modular Stereo

Microscope (Zeiss SteREO Discovery V12) to look into the sediment components. It is clear that abundant biogenic tests, especially foraminiferal tests, are observed during these sediment intervals (Figure A1). On the other hand, increased concentration of CaCO$_3$ is highly coherent with the abundance of planktic foraminifera species *G.ruber* and SST (Shi et al., 2014), indicating a substantial effect of SST on CaCO$_3$.

In the original manuscript, we made a mistake for LSR in Figure 3. In the revised manuscript, this issue has been corrected (Figure 3).

In addition, high sedimentation (>40-60 cm/ka) in the original manuscript mainly occurred during HS2 and HS3. This can be caused by uncertainties of age control points at 23.476 ka (DO2) and 29.995 ka (H3). In the revised manuscript, these two age control points have been eliminated from the Chinese Stalagmite tuned age model. Even with this more conservative tuning approach, the conclusions on sedimentary oxygenation variations remain the same as before and robust.

Lines 365-266: if U concentrations are affected by volcanic material over the last 8.5 yrs, then surely so are other sedimentary properties? I would like to see an argument in the main text discussing why certain proxy methods are deemed not to be influenced by this volcanic material, whilst others are. If it turns out that interval should not be used than this creates the complication of not being able to compare the down core data with more modern.

**Reply#17:** The occurrence of volcanic material has been confirmed by positive Eu anomalies and it has substantial effects on concentration of terrigenous materials (Zhu et al., 2015). This argument has been included in the revised text. Although the focus of this manuscript is not the ventilation changes during the Holocene, the Kuroshio Current does play a crucial role in controlling the ventilation of the OT during the Holocene. Please see **Reply#4**.

Line 370: change 'seems' to 'may'.
**Reply#18:** Revised. "seem to" was replaced by 'may". Line 376.

Lines 372-377: are there any other studies that use Mo/Mn ratios as a sedimentary oxygenation proxy, to support your interpretation?

**Reply#19:** To our knowledge, Mo/Mn ratios are not used as sedimentary oxygenation proxy in previous studies. In this study, we use both Mo/U ratio and excess U concentration to reconstruct sedimentary oxygenation changes. Among these, excess U concentration has been widely used for past sedimentary oxygenation changes (Jaccard and Galbraith, 2013; Jaccard et al., 2016; Jaccard et al., 2009). The strong positive correlation coefficient between excess U and Mo/Mn ratio in core CSH1 indicates its reliability and supports our interpretation. In contrast the ratio of Mo/Mn is easier to qualitatively assess indication of low vs. high oxygen environments, supported by the individual contents of Mn and Mo, respectively.

Line 380: define oxygen deficient.

**Reply#20:** . In the revised manuscript, "oxygen-deficient" was replaced by "suboxic". The sentence now reads as follows, Lines 387-388:

"......together with lower Mn concentrations suggest suboxic depositional conditions during the late deglacial period (15.8 ka–9.5 ka)......"

The oxygen thresholds was given in lines 396-398.

Discussion: Lines 387-392: I would recommend the authors to use more appropriate scheme that is used for sea-water that includes hypoxic, for example as defined by Bianchi et al. (2012). You will also found that suboxic is classified as < 2-10 μmol/l.

**Reply#21:** Thanks. The oxygen content scheme for seawater developed by Bianchi et al.(2012) has been adopted in the revised manuscript. Lines 396-398 in the revised MS. Following the Reviewer's suggestion we changed the sentence to:

" Here, we adopt the definition of oxygen thresholds by Bianchi et al. (2012) for oxic ($>120$ μmol/kg $O_2$), hypoxic ($<60–120$ μmol/kg $O_2$) and suboxic ($<2–10$ μmol/kg $O_2$) conditions, whereas anoxia is the absence of measurable oxygen."

Lines 403-405: I do not understand this reasoning. You have not linked weakly restricted basin settings with euxinia?

**Reply#22:** We thank the reviewer for this suggestion. The previous manuscript had inaccurate statement. Now, we changed this sentence to make it more explicit. The sentence was amended as follows. Lines 409-410 in the revised MS.

"Given that the northern OT is located in an open oceanic setting, we use these two proxies to evaluate the degree of oxygenation in sediments."

It is confusing talking about ppm in the main text, whilst Figure 4 gives concentrations in µg/g.

**Reply#23:** Corrected. In the revised manuscript, the uniform concentration unit, µg/g has been used for all trace elemental concentrations in the main text. Lines 411 and 417 in the revised MS.

Lines 406-412: Mo/U ratios are not shown in the manuscript. This is out of the blue.

**Reply#24:** Revised. Mo/U ratio has been included in new Figure 4m.

Lines 413-425: two studies. More importantly though, Figure 4 shows no benthic foraminifera data, and it is therefore impossible to confirm this claim of ventilation pattern from benthic foraminiferal assemblages to be similar to that of the RSEs. It would also be good to see a more critical discussion about this proxy.

**Reply#25:** We understand the point raised by the Reviewer. For this study, the benthic foraminiferal species were not counted in core CSH1. The benthic foraminiferal census data in cores E017 (1826 m water depth), 255 (1575 m water depth) are used to indicate the variations in ventilation of the middle and southern OT. The age model for core 255 (core length 655cm) was determined by two AMS $^{14}$C dates of *N. dutertrei* at depth 370 cm (9.17 ka) and 590 cm (18.8 ka) (Jian et al., 1996), whereas the age model for core E017 was established by six age control points (Li et al., 2005). Although the down-core abundance of hypoxia-affine species in both studies are similar to each other (Li et al., 2005), we here focus on the benthic foraminiferal census data from core E017.

We recalibrated the AMS $^{14}$C dates using the CALIB 7.04 software with the Marine 13 calibration dataset (Reimer et al., 2013) (ΔR=0) and compare the profiles of oxygen-like species and hypoxia-like species with our Mo/Mn and excess U. For the sake of simplicity, the abundance of *Bulimina aculeata* (hypoxia-indicating species) and *Cibicidoides hyalina* (oxygen-rich indicating species) have been included in new Figure 6 in the revised manuscript. High relative abundance of *B.aculeata* and low *C. hyalina* suggest the dominance of a hypoxic environment, whereas oxic conditions prevailed after ~7 ka. This is consistent with our RSE data, suggesting a widespread occurrence of oxygen-depleted water in the Okinawa Trough during the last deglaciation.

Lines 425-428: No. There is at least 800 meter water depth difference between your core and the others. The core of the current study is situated just above the low oxygen zone, whereas those of the other two studies are in /below the low oxygen zone.

**Reply#26:** Thanks for your comments. The seafloor bathymetry is much deeper in the southern OT, and shoals gradually toward the northern OT. Although our core is above the sill depth (1100 m), while others below (1100 m), previous investigations show the hydrographic characteristics in the mid-depth and deep OT are mainly regulated by the NPIW in the western boundary region of the Philippine Sea that flows into the OT through the Kerama Gap (1100 m) and the channel east of Taiwan (775 m) (Nakamura et al., 2013). Thus, ventilation signals recorded in these cores are mainly controlled by the same physical processes, though export productivity in different areas also exerts some impacts on deepwater and sedimentary oxygenation.

Lines 439-448: why are you looking at one NE Pacific to find out what is happening at your core site? There are several studies from across the Pacific that show something happening around the same time period (for example see Moffit et al., 2015), and Galbriath and Jaccard (2015), so rather than repeating the same discussion for a very small area, it would be easier to build up on those results.

**Reply#27:** At present, a contrasting distribution of dissolved oxygen concentration of the subsurface water can be observed in the eastern and western North Pacific margins (Figure 1) , which is characterized by strong Oxygen Minimum Zone in the eastern margin and oxic condition in the western margin. The benthic foraminiferal assemblages from ODP site 1017 exhibit a strengthening of the OMZ during warm periods and weakening during cold periods (Cannariato and Kennett, 1999). The question whether an expanded OMZ can extend toward the western NW Pacific remains elusive in the geological past during warming intervals. In fact, the key question involved is how to explain the cause of oxygenation variation on basin-wide scale. Comparison of our results from the western North Pacific with those of the eastern North Pacific aids to understand the mechanism behind sedimentary oxygen changes.

We have now added more information into the section 6.2 and all these references have been included in Lines 442-457.

"These processes have been invoked in previous studies to explain the deglacial Pacific-wide variations in oxygenation by either one or a combination of these factors (Galbraith and Jaccard, 2015; Moffitt et al., 2015; Praetorius et al., 2015). Our data also suggest drastic variations in sedimentary oxygenation over the last 50 ka. However, the mechanisms responsible for sedimentary oxygenation variations in the basin-wide OT and its connection with ventilation of the open North Pacific remain unclear. In order to place our core results in a wider regional context, we compare our proxy records of sedimentary oxygenation ($U_{excess}$ concentration and Mo/Mn ratio) and export productivity ($CaCO_3$) (Figures 6a, b, c) with abundance of *Pulleniatina obliquiloculata* (an indicator of Kuroshio strength) and sea surface temperature (Shi et al., 2014), bulk sedimentary nitrogen isotope (an indicator of denitrification) (Kao et al., 2008), benthic foraminifera $\delta^{13}C$ (a proxy for ventilation) in cores PN-3 and PC23A (Rella et al., 2012; Wahyudi and Minagawa, 1997), abundance of benthic foraminifera (an indicator of hypoxia) in core E017 (Li et al., 2005) and ODP Site 1017 (Cannariato and Kennett, 1999) (Figures 6d-k). "

Lines 454-457: no, at those high temperatures you would only get a reduction in O2 of ~ 3 for one degree warming, and 15 for a four degree warming (assuming no large salinity changes). Higher glacial salinity would cause less reduction in O2.

**Reply#28:** Corrected. The reference by Brewer and Peltzer (2016) has been included in the revised manuscript. Lines 463-465. The sentence was changed to:

"Based on thermal solubility effects, a hypothetical warming of 1°C would reduce oxygen concentrations by about 3.5 μmol/kg at water temperatures around 22°C (Brewer and Peltzer, 2016), ......"

Lines 457-458: sentence does not make sense.

**Reply#29:** We have removed the sentence in the revised manuscript.

Lines *848-846*(484-486?): does not make sense. How does subsurface water oxygen consumption lead to lower oxygen concentrations in deeper waters?

**Reply#30:** Thanks for your suggestion. We have removed the sentence in the revised manuscript. The replenishment of oxygen in deep water is controlled by both lateral advection and vertical supply. The oxygen consumption in subsurface water would reduce the oxygen supply, thus lead to a lower oxygen concentration in deeper waters.

Lines 491-494: again not taking into account other factors that influence CaCO3 accumulation and preservation in sediments. For discussion on Kuroshio Current: see Lim et al. (2017).

**Reply#31:** Please see **Reply #16** for $CaCO_3$.

Total Hg concentrations (Lim et al., 2017) has been invoked to explain variations in the intensity of Kuroshio. Interestingly, they concluded that the intensity of Kuroshio strengthened rapidly since 9.3 ka, whereas weakening and/or changing path of Kuroshio occurred during the last glaciation (20 ka - 9.3 ka). This conclusion further confirms weakening effect of Kuroshio on ventilation during the glacial period but an increase since 8.5 ka, consist with our inference. We have added Lines 528-531.

"More recently, lower hydrothermal total Hg concentration during 20 ka - 9.6 ka, associated with reduced intensity and/or variation in flow path of KC, relative to that of Holocene recorded in core KX12-3 (1423 water depth) (Lim et al., 2017), further validates our inference."

Line 544-550: coined? Matsumoto et al. (2002) discuss one radiocarbon age from the Santa Barbara basin in relation to oxygen content, but at no point do they propose that GNPIW was stronger oxygenated. Cartapanis et al. (2011) and Ohkushi et al. (2013) discuss that the NE OMZ Pacific strengthened and weakened at millennial time scales, not glacial interglacial timescales. Also around the equatorial Pacific it is suggested that there was no difference in intermediate water oxygenation between the last glacial and Holocene (Hoogakker et al., 2018). Further down in the South Pacific Lu et al. (2016) suggest that upper waters were depleted in oxygen during the last glacial period.

**Reply#32:**In Matsumoto et al. (2002), published in Quaternary Science Reviews (2002, 21, 1693-1704), they used a compilation of benthic foraminiferal $\delta^{13}C$ to reveal the deep hydrography of the North Pacific. On page 1700 of this paper, they stated that "Although a water mass that reaches 2000 m should not be called intermediate water in the sense of modern physical oceanography, here we will refer to it as glacial NPIW (GNPIW) for the lack of a better name."

According to our understanding, GNPIW refers to the nature of NPIW during cold intervals in the geological past, which is used to describe the state of NPIW at a variety of timescales, such as glacials, stadials, and Heinrich cold events. It should be noted that, since then, this term have been widely used in the literature related to NPIW (Cartapanis et al., 2011; Max et al., 2017; Worne et al., 2019).

Lines 550-556: generalised comment, what about brine (aka Kim et al. 2011).

**Reply#33:** Thanks. In the revised Manuscript, we shorten these sentences. Lines 562-563 in the revised MS. The sentence now was changed to:

"The intensified formation of GNPIW due to additional source region in the Bering Sea was proposed by Ohkushi et al. (2003) and Horikawa et al. (2010)."

Lines 556-559: what is intensified GNPIW?

**Reply#34:**"intensified GNPIW" means improved formation of NPIW. Here, we also use enhanced NPIW formation to replace intensified GNPIW. Lines 569-570 in the revised MS. The sentence was amended as follows.

"......validate such inference, suggesting pronounced effects of intensified NPIW formation in the OT."

Discussion lines 560-571: needs tightening, it is unclear where this goes and how it relates to this study?

**Reply#35:** We have rewrote these sentences. In the revised manuscript, the aim of this paragraph is to clarify the process of intensified GNPIW during HS1 and its substantial control on sedimentary oxygenation of the northern OT. Lines 571-586 in the revised MS. We have changed the sentences as follows:

"During HS1, a stronger formation of GNPIW was supported by proxy studies and numerical simulations. For example, on the basis of paired benthic-planktic (B-P) $^{14}$C data, enhanced penetration of NPIW into a much deeper water depth during HS1 relative to the Holocene has been revealed in several studies (Max et al., 2014; Okazaki et al., 2010; Sagawa and Ikehara, 2008), which was also simulated by several models (Chikamoto et al., 2012; Gong et al., 2019; Okazaki et al., 2010). On the other hand, increased intermediate water temperature in the subtropical Pacific recorded in core GH08-2004 (1166 m water depth) (Kubota et al., 2015) and young deep water observed in the northern South China Sea during HS1 (Wan and Jian, 2014) along downstream region of NPIW are also related to intensified NPIW formation. Furthermore, the pathway of GNPIW from numerical model simulations (Zheng et al., 2016) was similar to modern observations (You, 2003). Thus, all these evidence imply a persistent, cause and effect relation between GNPIW ventilation, the intermediate and deep water oxygen concentration in the OT and sediment redox state during HS1. In addition, our RSEs data also suggested a similarly enhanced ventilation in HS2 (Figures 6b and c) that is also attributed to intensified GNPIW formation."

Line 629: what is ocean ventilation seesaw? There is hardly any explanation for this in the main text, and Figure 7 shows strength of AMOC in the Atlantic and compares this with the current study.

**Reply#36:** In the revised manuscript, the "ocean ventilation seesaw" has been replaced by "the mechanism behind such out-of-phase pattern between the ventilation in the subtropical North Pacific and the North Atlantic deep water formation" Lines 647-649 in the revised MS.

On the other hand, we also have added some discussion about the North Atlantic Climate in the 1st paragraph of section 6.3 and modified the text of the following sections to make it more logical.

Lines 619-630

"One of the characteristic climate features in the Northern Hemisphere, in particular the North Atlantic is millennial-scale oscillation during glacial and deglacial periods. These abrupt climatic events have been widely thought to be closely related to varying strength of Atlantic Meridional Overturning Circulation (AMOC) (Lynch-Stieglitz, 2017). One of dynamic proxies of ocean circulation, $^{231}Pa/^{230}Th$ reveals that severe weakening of AMOC only existed during Heinrich stadials due to increased freshwater discharges into the North Atlantic (Böhm et al., 2015; McManus et al., 2004). On the other hand, several mechanisms, such as sudden termination of freshwater input (Liu et al., 2009), atmospheric $CO_2$ concentration (Zhang et al., 2017), enhanced advection of salt (Barker et al., 2010) and changes in background climate (Knorr and Lohmann, 2007) were proposed to explain the reinvigoration of AMOC during the B/A."

Lines 631-632

"Our RSEs data in the Northern OT and endobenthic $\delta^{13}C$ in the Bering Sea (Figures 7a-c) both......"

Lines 647-649

"However, the mechanism behind such out-of-phase pattern between the ventilation in the subtropical North Pacific and the North Atlantic deep water formation remains unclear."

Captions

Figure A1 Photomicrographs with Modular Stereo Microscope (Zeiss SteREO Discovery V12) show that both detrital and biogenic components of sediment coarse fraction (>63 μm) for 8.23 ka (120-124 cm), 9.26 ka (144-148 cm), 10.98 ka (184-188 cm), 11.66 ka (200-204 cm), 12.92 ka (232-236 cm), 14.05 ka (264-268 cm), and 15.18 ka (296-300 cm) in core CSH1 at 200 X magnification.

Figure 3. (a) Lithology and oxygen isotope ($\delta^{18}O$) profile of planktic foraminifera species *Globigerinoides ruber* (*G.ruber*) in core CSH1. (b) Plot of ages versus depth for core CSH1. Three known ash layers are indicated by solid red rectangles. (c) Time series of linear sedimentation rate (LSR) from core CSH1. (d) Comparison of age model of core CSH1 with Chinese Stalagmite composite $\delta^{18}O$ curve of (Cheng et al., 2016). Tie points for CSH1 core chronology (Table 2) in Figures 3c and d are designated by colored crosses.

Figure 6. Proxy-related reconstructions of mid-depth sedimentary oxygenation at site CSH1 (this study) compared with oxygenation records from other locations of the North Pacific and published climatic and environmental records from the Okinawa Trough.

**Figure A1**

[Figure]

Figure 3

[Figure]

Figure 6

[Figure]

**Point-by-point response to referee comments**

**Response to reviewer**

Key:

- Reviewers' comments

- Author's response

- Modified text in the manuscript

Anonymous Referee #2

Zou et al., present a rather interesting study focusing on reconstructing the oxygenation history in the Okinawa Trough covering the last 50 kyrs. Specifically, the authors attempt to disentangle the typically confounding influence of export production (and by inference the oxygen consumption related to organic matter degradation) and bottom water ventilation on the sedimentary redox condition. Their geochemical records largely corroborate previous findings in that oxygenation patterns at intermediate depths in the North Pacific were primarily controlled by the production and ventilation of North Pacific Intermediate Water. Specifically, conditions were generally better oxygenated during stadials, when NPIW was generally better ventilated and vertically expanded. Furthermore, their data support a general expansion of oxygen-depleted waters at intermediate depths during the B/A occupying large swaths of the North Pacific.

The manuscript is well documented and quite detailed in places. The argumentation could be somewhat streamlined (see comments below) and would certainly benefit from editorial support. I would also recommend the argumentation to focus on the aspects outlined in the title and abstract.

**Reply#1:** We thank the reviewer for the recognition of significance of this study. Based on the suggestions of the reviewer, we revised our manuscript thoroughly with a focus on sedimentary oxygenation changes in the subtropical North Pacific and its linkages to North Atlantic Climate. In the following, we address each specific point raised by the reviewer.

l. 35 – deep ocean carbon sequestration is certainly one the reasons potentially explaining lower glacial pCO2 concentrations, but certainly not the only one. Please rephrase to avoid unnecessary confusion

**Reply#2:** Thanks for your suggestion. We rephrased this sentence as the following. Lines 32-34 in the revised MS.

"The deep ocean carbon cycle, especially carbon sequestration and outgassing, is one of the mechanisms to explain variations in atmospheric CO2 concentrations on millennial and orbital timescales."

l. 36 – I would suggest rephrasing as follows – However, the potential role of subtropical North Pacific subsurface waters in modulating.

**Reply#3:** Agreed and we have done so. Thanks. Lines 34-36. The sentence was amended as follows.

"However, the potential role of subtropical North Pacific subsurface waters in modulating atmospheric $CO_2$ levels on millennial timescales is poorly constrained."

l. 48 and throughout – why is HS1 so much different from HS2 when considering their respective oxygenation history?

**Reply#4:** During HS1 and HS2, our RSEs suggest enhanced sedimentary oxygenation. As suggested by the Reviewer, a slight difference in the structure of ventilation mode can be observed (Figure 7). We think this slight discrepancy could be related to different climatic background of HS1 and HS2. Records from paleoclimate archives, such as ice core and Chinese cave stalagmites, show some differences in structure, duration, amplitude between HS1 and HS2. Such differences are thought to be related to the climate background state, such as, $CO_2$ concentration, ice sheet volume, AMOC intensity, sea ice extent and source of freshwater, etc. (Flückiger et al., 2006; Hemming, 2004; Kaspi et al., 2004; Lynch-Stieglitz et al., 2014) . The response of NPIW to HS1 and HS2 events could be different, and thus cause a slight difference in sedimentary oxygenation. On the other hand, the discrepancy in export productivity ($CaCO_3$) during these two cold intervals in core

CSH1 could also play a role in controlling sedimentary oxygenation.

l. 62 – agreed. But I would add that in addition of the flushing of a poorly ventilated deep water mass upon the resumption of NADW, many export production records show a drastic increase during the B/A (e.g. Kohfeld& Chase, 2011), which could account for enhance oxygen removal associated with organic matter respiration upstream of the core site location.

**Reply#5:** Agreed with your suggestion and revised. Lines 51-52. The sentence was changed to

"......due to decreased NPIW formation and enhanced export production......"

l. 70 – AT the sediment-water interface

**Reply#6:** We reworded the sentence. Lines 61-66 in revised Manuscript. It was amended as follows.

"A more sluggish deep ocean ventilation combined with a more efficient biological pump widely thought to facilitate enhanced carbon sequestration in the ocean interior, leading to atmospheric $CO_2$ drawdown during glacial cold periods (Sigman and Boyle, 2000). These changes are tightly coupled to bottom water oxygenation and sedimentary redox changes on both millennial and orbital timescales (Hoogakker et al., 2015; Jaccard and Galbraith, 2012; Sigman and Boyle, 2000)."

l. 83 . . . in marine sediment cores

**Reply#7:** Corrected. Line 76. The sentence was amended as follows.

"......in marine sediment cores."

l. 86 . . . in the subarctic North Pacific.

**Reply#8:** Corrected. Line 78. The sentence was amended as follows.

"...... last glaciation and Holocene in the subarctic North Pacific."

l. 92 – I would suggest to briefly explain what cabbelingmeans

**Reply#9:** Done. Cabbeling is a mixing process to form a new water mass with increased density than that of parent water masses. We have added in Lines 85-86.

"......that cabbeling, a mixing process to form a new water mass with increased density than that of parent water masses, is the principle mechanism responsible......"

l. 96 and throughout (incl. Fig. 6) – this may well be a semantic issue, but benthic d13C cannot be considered as a ventilation proxy per se, as the isotopic value can be obfuscated by air-sea gas exchange in locations where subsurface water masses form.

**Reply#10:** We agree with the comment by the Reviewer that benthic $\delta^{13}C$ changes are influenced by a variety of factors, such as air-sea equilibration, ocean-circulation changes, and productivity changes, etc. Previous studies also revealed benthic $\delta^{13}C$ patterns at basin-wide scales can reflect ocean-circulation changes (Charles and Fairbanks, 1992; Charles et al., 1996; Ninnemann and Charles, 2002). In this study we also noticed similar trends in benthic $\delta^{13}C$ during ~22 ka and ~14 ka between cores PN-3 (Okinawa Trough) and PC23A (Bering Sea) (Figure 4), despite their great distance. In the revised manuscript, we add additional information for benthic $\delta^{13}C$. The sentence was amended Lines 89-92.

"Benthic foraminiferal $\delta^{13}C$, a quasi-conservative tracer for water mass, from the North Pacific suggested an enhanced ventilation (higher $\delta^{13}C$) at water depths of < 2000 m during the last glacial period (Keigwin, 1998; Matsumoto et al., 2002)."

l. 151 – processes that are related to the supply of oxygen by ocean circulation and. . .

**Reply#11:** Revised. Lines 150-153. The sentence was changed to:

"The sedimentary redox conditions are governed by the rate of oxygen supply from the overlying bottom water and the rate of oxygen removal from pore water (Jaccard et al., 2016), processes that are related to the supply of oxygen by ocean circulation and organic matter respiration, respectively."

l. 163 – technically one should specify that under reducing conditions it is the authigenic fraction of Mn (as opposed to its detrital background) that remains low.

**Reply#12:** Revised. Lines 162-163. The sentence was amended as follows:

"Under reducing conditions, the authigenic fraction of Mn (as opposed to its detrital background) ......"

l. 168 – please add adequate reference

**Reply#13:** The reference (Morford and Emerson 1999) has been included in the revised manuscript. Line 170.

l. 195 – volcanism

**Reply#14:** Revised. Line 198. "volcanic " has been changed to "volcanism".

§3 – I would suggest to substantially shorten this paragraph as the general oceanographic setting is already outlined in the introduction. I would recommend focusing on the aspects directly relevant to the argumentation (nutrient, dissolved O2).

**Reply#15:** Thanks for your suggestion. In the revised manuscript, we removed the 1$^{st}$ paragraph of section 3 and then we reworded some sentences of previous 3$^{rd}$ paragraph of section 3. Lines 218-227. The paragraph was amended as follows.

"A lower sea surface salinity (SSS) zone in summer relative to the one in winter in the ECS migrates toward the east of OT, indicating enhanced impact of the Changjiang discharge associated with summer EAM (Figures 2a and b). An estimated ~80% of the mean annual discharge of the river Changjiang is supplied to the ECS (Ichikawa and Beardsley, 2002) and in situ observational data show a pronounced negative correlation between the Changjiang discharge and SSS in July (Delcroix and Murtugudde, 2002). Consistently, previous studies from the OT reported such close relationship between summer EAM and SSS back to the late Pleistocene (Chang et al., 2009; Clemens et al., 2018; Kubota et al., 2010; Sun et al., 2005)."

l. 287 – why is the sedimentation rate so high during HS2 when both export production and detrital input (based on Al) are low? I would suggest verifying the age pointers for that interval.

**Reply#16:** Thanks for your suggestion and we have verified the age control points (Figure 3 and Table 2). In the original manuscript, higher sedimentation rate mainly occurred during HS2 and HS3 (> 40-60 cm/ka) and is mainly caused by uncertainties of age control points at 23.476 ka (DO2) and 29.995 ka (H3). In the revised manuscript, these two age control points have been eliminated. Even with this more conservative tuning approach, the conclusions remain the same as before and robust.

l.291 – 85 samples covering 50 kyrs cannot provide an average time resolution of 200 yrs.

**Reply#17:** We thank the reviewer to point out this mistake. Now we have corrected it to ~ 600 years. Lines 286-287 in revised manuscript. The sentence was changed to:

"...... representing a temporal resolution of about 600 years (every 4 cm interval)......"

l. 345-346 – Maybe. But it may also suggest that the sedimentary CaCO3 content could be directly controlled by dilution. I would interpret the export productivity records with caution.

**Reply#18:** We thank the reviewer for the very helpful comments and suggestions. In the revised manuscript, we have ruled out the effects of some factors, such as dilution and dissolution, on $CaCO_3$. On the other hand, we have reworded this section to cautiously interpret $CaCO_3$ as a reliable proxy for export production at core CSH1. We have added the following information to Lines 339-358.

"Several lines of evidence support $CaCO_3$ as a reliable productivity proxy, particularly during the last deglaciation. The strong negative correlation coefficient (r = – 0.85, p<0.01) between Al and $CaCO_3$ in sediments throughout core CSH1 confirms the biogenic origin of $CaCO_3$ against terrigenous Al (Figure 4f). Generally, terrigenous dilution decreases the concentrations of $CaCO_3$. An inconsistent relationship between $CaCO_3$ contents and sedimentation rates indicates a minor effect of dilution on $CaCO_3$. Furthermore, the increasing trend in $CaCO_3$ associated with high sedimentation rate during the last deglacial interval indicates a substantial increase in export productivity (Figures 4a and d). The high coherence between $CaCO_3$ content and alkenone-derived sea surface water (SST) (Shi et al., 2014) indicates a direct control on $CaCO_3$ by SST. Moreover, a detailed comparison between $CaCO_3$ concentrations and the previously published foraminiferal fragmentation ratio (Wu et al., 2004) shows, apart from a small portion within the LGM, no clear co-variation between them. These evidence suggest that CaCO$_3$ changes are driven primarily by variations in carbonate primary production, and not overprinted by secondary processes, such as carbonate dissolution through changes in the lysocline depth and dilution by terrigenous materials. Likewise, a similar deglacial trend in CaCO$_3$ is also observed in core MD01-2404 (Chang et al., 2009), indicating a ubiquitous, not local picture in the OT. All these lines of evidence thus support CaCO$_3$ of core CSH1 as a reliable productivity proxy to a first order approximation."

l.371-372 – interesting idea!

**Reply#19:** Thanks.

l. 427 (**438?**) – supply seems more adequate than provision.

**Reply#20:** Agreed and revised. Line 440. "provision" was replaced by "supply".

l. 487-488 – please also consider citing Galbraith et al., 2007.

**Reply#21:** The paper has been included in the revised manuscript. Line 496.

l. 489-490 – please keep in mind that O2 can be consumed upstream of the core site location as the removal of O2 in relation to organic matter degradation is integrated over the flow path of a give subsurface water mass.

**Reply#22:** Thanks for your suggestion. We agree with the comment.

Fig. 6C – shouldn't the grey triangle right of the vertical axis be flipped upside down (i.e. high Mo/Mn coincident with low oxygenation)?

**Reply#23:** Thanks. High Mo/Mn ratio indicates low oxygen condition. The sign in Fig. 6c has been corrected.

[revised manuscript text omitted]


Fig.1

[Figure]

[Figure]

Fig.2

[Figure]

Fig.3

[Figure]

.

Fig.4

[Figure]

[Figure]

Fig.5

[Figure]

[Figure]

[Figure]

Fig.7

[Figure]

[Figure]

---

## Author Response (AR2)

**Point-by-point response to referee comments**

**Response to reviewers**

Key:

- Reviewers' comments

- Author's response

- Modified text in the manuscript

Editor Decision:

Comments to the Author:

Dear Jianjun Zou,

I have now received feedback from the two reviewers on your revised manuscript "Millennial-scale variations of sedimentary oxygenation in the western subtropical North Pacific and its links to the North Atlantic climate" submitted to Climate of the past. Both reviewers are very pleased with the way you addressed and responded to their comments given in the first round of reviews, and recommendto accept the paper after technical corrections. I will therefore ask you to respond to the minor comments raised in the new reviewer reports before final acceptance of the paper.

Best regards,

BjørgRisebrobakken

Editor, Climate of the Past

Reply#1: We thank both reviewers for commenting positively on the revised version of our original manuscript. We particularly appreciate that each reviewer found that our manuscript is worth for publication after minor technical corrections.

Referee #1

Zou et al have made substantial revision to their manuscript 'Millennial-scale variations of sedimentary oxygenation in the western subtropical North Pacific and its links to North Atlantic climate'.

I only have a few (small) comments, and I am happy to see the manuscript published once these have been considered:

Abstract:

Lines 36-38: the way I read this is that an increase in deep ocean respired $CO_2$

concentrations leads to deoxygenation of the upper water column (e.g. top 1000 m represents subsurface layer). Subsurface is not necessarily the same as mid-depth, which I think it what you actually mean? Perhaps remnant of (copied and pasted from response reviewers comments):

'Lines 848-846(484-486?): does not make sense. How does subsurface water oxygen consumption lead to lower oxygen concentrations in deeper waters?

Reply#30: Thanks for your suggestion. We have removed the sentence in the revised manuscript. The replenishment of oxygen in deep water is controlled by both lateral advection and vertical supply. The oxygen consumption in subsurface water would reduce the oxygen content in the ocean interior, thus lead to a lower oxygen concentration in deeper waters.'

Reply#2: Thanks for this comment. Here we would like to highlight the deoxygenation in the ocean interior at that time. In order to make a clear statement, the subsurface was replaced by ocean interior in Line 38 of the revised manuscript.

As stated in Reply#30 of the first-round of response letter, we suggested the oxygen consumption because of organic matter degradation in the ocean interior would reduce the vertical supply of oxygen to deeper waters, despite the replenishment of oxygen due to lateral advection.

Lines 45 to 48: perhaps rephrase; e.g. cold spells enhanced oxygenation linked to NPIW, while interglacial increase after 8.5 ka linked to intensification of Kuroshio Current.

Add a line about why the Kuroshio Current intensified?

Reply#3: Thanks. We rephrased this sentence as follows. Please refer to Lines 45-48 in the revised MS.

"The enhanced oxygenation during cold spells is linked to the North Pacific Intermediate Water (NPIW), while interglacial increase after 8.5 ka is linked to an intensification of the Kuroshio Current due to strengthened northeast trade winds over the tropics."

Line 62: add 'is' between pump and widely.

Reply#4: Added in Line 62 of the revised manuscript.

Introduction:

Lines 88-89: do you mean a small subpolar input or a minimum input, or reducing the input (as it read now)?

Reply#5: Thanks. The word lower was replaced by small in Line 88 of the revised manuscript.

Material and methods: could you add the response to this query to the section? Gives strength to your interpretation.

Reply#6: Thanks. We agree with this suggestion. In the revised manuscript (Lines 281-285), the following sentence (including references) has been added to give more information on sedimentation rate.

"Variation in sedimentation rate has been attributed to changes in eustatic sea level, summer EAM intensity, path and/or intensity of Kuroshio Current. Generally, sea level is thought to be the first-order factor for controlling linear sedimentation rate changes (Beny et al., 2018; Li et al., 2015; Zhao et al., 2017)."

Anonymous Referee #2

The authors have substantially revised their manuscript, which I feel now has the attributes to be accepted for publication in Climate of the Past. I appreciate the thoroughness of their rebuttal letter, well done!

I have two last very minor comments, which I would like to see addressed before the manuscript can be formally accepted - l. 311 and throughout – I'm not sure I understand why Mn is plotted as total Mn and not, like U and Mo, as Mnexcess or Mn/Al for example?

Reply#7: In this study, the Mo/Mn ratio is a ratio of total Mo concentration to total Mn concentration. The main reasons that the Mo/Mn ratio was used are ascribed to (1) the contrasting geochemical behaviors between these two elements and (2) a strong positive correlation between Mo/Mn and excess U.

The sedimentary Mn concentration in core CSH1 ranges between 0.04% and 0.06% with an average of 0.05% over the last 50ka, which is less than the Mn concentrations in Upper Crust (0.06%) (Taylor and McLennan, 2009), the suspended particulate matter of Yangtze River (0.116%) (Gaillardet et al., 1999) and the average shale (0.085%) (Li and Schoonmaker, 2014). Therefore, there is no excess Mn in sediments of core CSH1. The reason can be ascribed to dilution by biogenic materials and dissolution of Mn oxides and oxyhydroxides.

We also calculated the ratio of Mn/Al and it shows a similar downcore trend to that of total Mn concentration (Figure S1) and therefore we preferred the total Mn rather than Mn/Al ratio in the manuscript.

[Figure]

Figure S1 Downcore records of Mn concentration and Mn/Al ratio in core CSH1 over the last 50 ka.

Fig. 7, pannel d, the Pa/Th record has superposed measurements, which I think must be a small mistake.

Reply#8: In Fig.7d, the data of Pa/Th are taken from two references (Böhm et al., 2015; McManus et al., 2004). In order to make distinction of these two sources, different colored lines are used in revised Figure 7d.

[Figure]

**Figure 7.** Proxy records favoring the existence of out-of-phase connections between the subtropical North Pacific and North Atlantic during the last deglaciation and enhanced carbon storage at mid-depth waters. (a) U $_{excess}$ concentration in core CSH1; (b) Mo/Mn ratio in core CSH1; (c) benthic $\delta^{13}C$ record in core PC-23A in the Bering Sea (Rella et al., 2012); (d) Indicator of strength of Atlantic Meridional Ocean Circulation ($^{231}Pa/^{230}Th$) (Böhm et al., 2015; McManus et al., 2004); (e) Atmospheric $CO_2$ concentration (Marcott et al., 2014). Light gray and dark gray vertical bars are the same as those in Figure 4.

[revised manuscript text omitted]

Fig.2

[Figure]

Fig.3

[Figure]

Fig.4

[Figure]

Fig.5

[Figure]

Fig.6

[Figure]

Fig.7

[Figure]

